# STING promotes NLRP3 localization in ER and facilitates NLRP3 deubiquitination to activate the inflammasome upon HSV-1 infection

**Wenbiao Wang[1]ᵒ, Dingwen Hu[2]ᵒ, Caifeng Wu[1], Yuqian Feng[1], Aixin Li[2], Weiyong Liu[2], Yingchong Wang[2], Keli Chen[2], Mingfu Tian[2], Feng Xiao[2], Qi Zhang[2], Muhammad Adnan Shereen[2], Weijie Chen[1], Pan Pan[1], Pin Wan[1], Kailang Wu[2]\*, Jianguo Wu[1,2]\***

**1** Guangdong Key Laboratory of Virology, Institute of Medical Microbiology, Jinan University, Guangzhou, China, **2** State Key Laboratory of Virology, College of Life Sciences, Wuhan University, Wuhan, China

ᵒ These authors contributed equally to this work.
\* wukailang@whu.edu.cn (KW); jwu@whu.edu.cn (JW)

## Abstract

One of the fundamental reactions of the innate immune responses to pathogen infection is the release of pro-inflammatory cytokines, including IL-1β, processed by the NLRP3 inflammasome. The stimulator of interferon genes (STING) has the essential roles in innate immune response against pathogen infections. Here we reveal a distinct mechanism by which STING regulates the NLRP3 inflammasome activation, IL-1β secretion, and inflammatory responses in human cell lines, mice primary cells, and mice. Interestingly, upon HSV-1 infection and cytosolic DNA stimulation, STING binds to NLRP3 and promotes the inflammasome activation through two approaches. First, STING recruits NLRP3 and facilitates NLRP3 localization in the endoplasmic reticulum, thereby facilitating the inflammasome formation. Second, STING interacts with NLRP3 and attenuates K48- and K63-linked polyubiquitination of NLRP3, thereby promoting the inflammasome activation. Collectively, we demonstrate that the cGAS-STING-NLRP3 signaling is essential for host defense against HSV-1 infection.

## Author summary

The innate immune system is a primary host defense strategy to suppress the pathogen infections. One of the fundamental reactions of the innate immunity is the release of pro-inflammatory cytokines, including interleukine-1β (IL-1β), regulated by inflammasomes. The best-characterized inflammasomes is the NLRP3 inflammasome. STING has the essential roles in innate immune response against pathogen infections and is required for pathogen-induced inflammasome activation and IL-1β secretion. This study explores how STING regulates the NLRP3 inflammasome and reveals a distinct mechanism underlying such regulation upon herpes simplex virus type 1 (HSV-1) infection and cytosolic DNA stimulation. The authors propose that the cGAS-STING-NLRP3 axis is essential for host defense against HSV-1 infection.

**Data Availability Statement:** All relevant data are within the manuscript.

**Funding:** This work was supported by the National Natural Science Foundation of China (81730061 and 81471942), the National Health and Family Planning Commission of China (National Mega Project on Major Infectious Disease Prevention) (2017ZX10103005 and 2017ZX10202201), Guangdong Province "Pearl River Talent Plan" Innovation and Entrepreneurship Team Project (2017ZT07Y580), and Postdoctoral Science Foundation of China (2018T110923). The funders had no role in study design, data collection and analysis, decision to publish, or preparation of the manuscript.

**Competing interests:** The authors have declared that no competing interests exist.

## Introduction

The innate immune system detecting pathogens through recognition of molecular patterns is a primary host defense strategy to suppress the infections [1]. Recognition of pathogens stimuli, known as pathogen-associate molecular patterns (PAMPS), is relied on pattern recognition receptors (PRRs). Several families of PRRs have been described, including the Toll-like receptor (TLR) [2], RIG-I-like receptor (RLR) [3], NOD-like receptor (NLR) [4], and C-type lectin receptor (CLR) [5]. The NLRs involved in the assembly of large protein complexes referred to as inflammasomes are emerging as a major route by which the innate immune system responds to pathogen infections [6]. One of the fundamental reactions of the innate immunity is the procession and release of pro-inflammatory cytokines, including interleukine-1β (IL-1β), a pleiotropic cytokine playing crucial roles in inflammatory responses in addition to instructing immune responses [7]. The best-characterized inflammasomes is the NLRP3 inflammasome, which consists of three major components: a cytoplasmic sensor NLRP3 (NACHT, LRR and PYD domains-containing protein 3), an adaptor ASC (apoptosis-associated speck-like protein with CARD domain), and an interleukin-1β-converting enzyme pro-Caspase-1 (cysteinyl aspartate-specific proteinase-1) [6]. NLRP3 and ASC together promote the cleavage of pro-Casp-1 to generate active subunits p20 and p10, which regulate IL-1β maturation [8].

The stimulator of interferon genes (STING) has the essential roles in innate immune response against pathogen infections [9]. Upon binding of cytoplasmic DNA, cyclic GMP-AMP synthase (cGAS) catalyzes the formation of cyclic guanosine monophosphate-adenosine monophosphate (cGAMP) by binding to STING. STING subsequently co-localizes with TBK1 and IRF3, leading to induction of type I IFNs, and recruits TRAF6 and TBK1 or TRAF3 and IKKα to activate the NF-κB pathway [10, 11]. In human myeloid cells, STING is involved in cytosolic DNA induced-NLRP3 inflammasome activation [12], and in mice BMDMs, STING is required for pathogen-induced inflammasome activation and IL-1β secretion [13, 14].

We explored how STING regulates the NLRP3 inflammasome and reveal a distinct mechanism underlying such regulation upon herpes simplex virus type 1 (HSV-1) infection and cytosolic DNA stimulation. HSV-1 causes various mild clinical symptoms, while in immunocompromized and neonates individuals, it can cause herpes simplex encephalitis, which may lead to death or result in some neurological problems [15]. But the detailed mechanisms by which HSV-1 regulates the NLRP3 inflammasome are largely unknown. We demonstrate that HSV-1-induced NLRP3 inflammasome activation is dependent on the cGAS-STING pathway. STING recruits and improves NLRP3 localization in the endoplasmic reticulum, and binds and removes NLRP3 polyubiquitination, thereby promoting the inflammasome activation. We propose that the cGAS- STING-NLRP3 axis is essential for host defense against HSV-1 infection.

## Results

### STING interacts with NLRP3 to facilitate the inflammasome activation

We initially determined the correlation between STING and NLRP3, and showed that STING and NLRP3 interacted with each other in human embryonic kidney (HEK293T) cells (Fig 1A and 1B). The NLRP3 inflammasome consists of three major components, NLRP3, ASC, and pro-Casp-1 [6]. We explored whether STING interacts with ASC and/or pro-Casp-1, and clearly revealed that STING interacted with NLRP3, but not with ASC or pro-Casp-1 (Fig 1C). NLRP3 protein harbors several prototypic domains, including PYRIN domain (PYD),

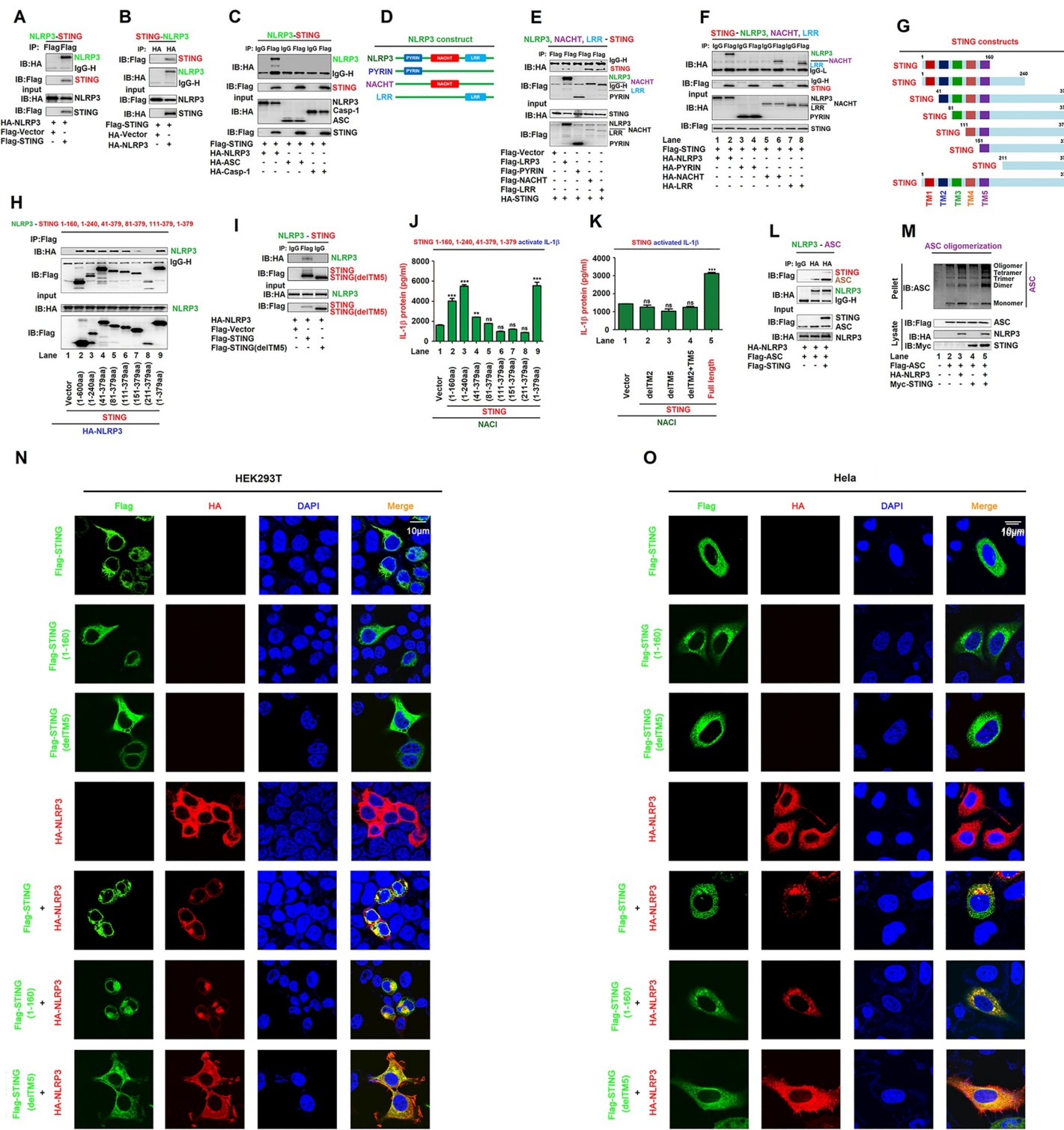

**Fig 1. STING interacts with NLRP3 to facilitate the inflammasome activation.** (**A–C**) HEK293T cells were co-transfected with pFlag-STING and pHA-NLRP3 (A and B), or with pFlag-STING, pHA-NLRP3, pHA-ASC, and pHA-Casp-1 (C). (**D**) Diagrams of NLRP3, PYRIN, NACHT, and LRR. (**E, F**) HEK293T cells were co-transfected with pHA-STING and pFlag-NLRP3, pFlag-PYRIN, pFlag-NACHT, or pFlag-LRR (E), or with pFlag-STING and pHA-NLRP3, pHA-PYRIN, pHA-NACHT, or pHA-LRR (F). (**G**) Diagrams of STING and its truncated proteins. (**H, I**) HEK293T cells were co-transfected with pHA-NLRP3 and pFlag-STING, truncated proteins (H) or TM5-deleted STING(delTM5) (I). (**J, K**) HEK293T cells were co-transfected with plasmids encoding NLRP3, ASC, pro-Casp-1, and pro-IL-1β to generate a pro-IL-1β cleavage system (NACI), and transfected with pFlag-STING, truncated proteins (J), TM2-deleted STING(delTM2), TM5-deleted STING(delTM5) or TM2 and TM5-deleted STING(delTM2 and 5) (K). IL-1β in supernatants was determined by ELISA. (**L**) HEK293T cells were co-transfected with pFlag-ASC, pHA-NLRP3, or pFlag-STING. (**A–C, E, F, H, I,** and **L**) Cell lysates were subjected to co-immunoprecipitation (Co-IP) using IgG (Mouse) and anti-Flag antibody (C, F), IgG (Rabbit) and

anti-HA antibody (L), anti-HA antibody (B), and anti-Flag antibody (A, E, H and I), and analyzed by immunoblotting using anti-HA or anti-Flag antibody (top) or subjected directly to Western blot using anti-Flag or anti-HA antibody (input) (bottom). (**M**) HEK293T cells were co-transfected with pFlag-ASC, pHA-NLRP3, and pMyc-STING. Pellets were subjected to ASC oligomerization assays (top) and lysates were prepared for Western blots (bottom). (**N, O**) HEK293T cells (N) or Hela cells (O) were co-transfected with pFlag-STING and/or pHA-NLRP3. Sub-cellular localization of Flag-STING (green), HA-NLRP3 (red), and DAPI (blue) were examined by confocal microscopy. Data shown are means ± SEMs; **p < 0.01, ***p < 0.0001; ns, no significance.

NACHT-associated domain (NAD), and Leucine rich repeats (LRR) [16]. Next, the domain of NLRP3 involved in the interaction with STING was determined by evaluating the plasmids encoding NLRP3, PYRIN, NACHT, or LRR (Fig 1D) as described previously [17]. Like NLRP3, NACHT and LRR interacted with STING, but PYRIN failed to interact with STING (Fig 1E), and consistently, STING interacted with NLRP3, NACHT, and LRR (Fig 1F, lanes 2, 6 and 8), but not with PYRIN (Fig 1F, lane 4). In another hand, STING comprises five putative transmembrane (TM) regions [18]. The domain of STING required for the interaction with NLRP3 was assessed by analyzing plasmids encoding wild-type (WT) STING and seven truncated proteins (Fig 1G). Like WT STING(1–379aa) (Fig 1H, lane 9), the truncated proteins STING(1–160aa), STING(1–240aa), STING(41–379aa), STING(81–379aa), and STING(111–379aa) interacted strongly with NLRP3 (Fig 1H, lane 2–6), STING(151–379aa) associated weakly with NLRP3 (Fig 1H, lane 7), but STING(211–379aa) failed to interact with NLRP3 (Fig 1H, lane 8), indicating that TM5 (151–160aa) of STING is involved in the interaction with NLRP3. We also demonstrated that deleted-TM5 STING failed to interact with NLRP3 (Fig 1I). Taken together, these results suggest that TM5 (151–160aa) of STING is involved in the interaction with NLRP3.

The role of STING in the regulation of the NLRP3 inflammasome was explored by using a pro-Casp-1 activation and pro-IL-1β cleavage cell system as established previously [19]. In this system (NACI), HEK293T cells were co-transfected with plasmids encoding NLRP3, ASC, pro-Casp1, and pro-IL-1β. In the NACI cells, IL-1β secretion was stimulated by STING(1–160aa), STING(1–240aa), STING(41–379aa), or STING (Fig 1J, lane 2–4 and 9), but not by STING(81–379aa), STING(111–379aa), STING(151–379aa) or STING(211–379aa) (Fig 1J, lane 5–8), suggesting that TM2 (41–81aa) of STING is required for the induction of IL-1β secretion. In addition, IL-1β secretion was stimulated by STING, but not by deleted-TM2 STING, deleted-TM5 STING, or TM2 and TM5-deleted STING(delTM2 and 5) (Fig 1K), indicating that both TM2(41–81aa) and TM5(151–160aa) are required for the induction of NLRP3 inflammasome activation. Notably, STING promoted the NLRP3-ASC interaction (Fig 1L), an indicator of the inflammasome assembly [20] and enhanced NLRP3-mediated ASC oligomerization (Fig 1M), which is critical for inflammasome activation [21]. Moreover, in HEK293T cells (Fig 1N) and HeLa cells (Fig 1O), NLRP3 and STING co-localized and formed large spots in the cytosol (Fig 1N and 1O), an indication of the NLRP3 inflammasome formation [22]. Taken together, STING interacts with NLRP3 through TM5 domain and promotes the assembly and activation of the NLRP3 inflammasome through TM2 domain.

## HSV-1 infection promotes the STING-NLRP3 interaction

STING plays a key role in host innate immune response upon pathogen infections and cytosolic DNA simulation [18]. We evaluated the effects of HSV-1 infection and HSV120 transfection, a biotinylated dsDNA representing the genomes of HSV-1 that efficiently induces STING-dependent type I IFN production as reported previously [23], on the STING-NLRP3 interaction. In TPA-differentiated human leukemic monocyte (THP-1) macrophages, HSV-1 infection facilitated endogenous NLRP3-STING interaction and promoted endogenous STING-NLRP3 interaction (Fig 2A and 2B). In HEK293T cells and HeLa cells (Fig 2D), the STING-NLRP3 interaction was enhanced upon HSV-1 infection (Fig 2C and 2D). Similarly,

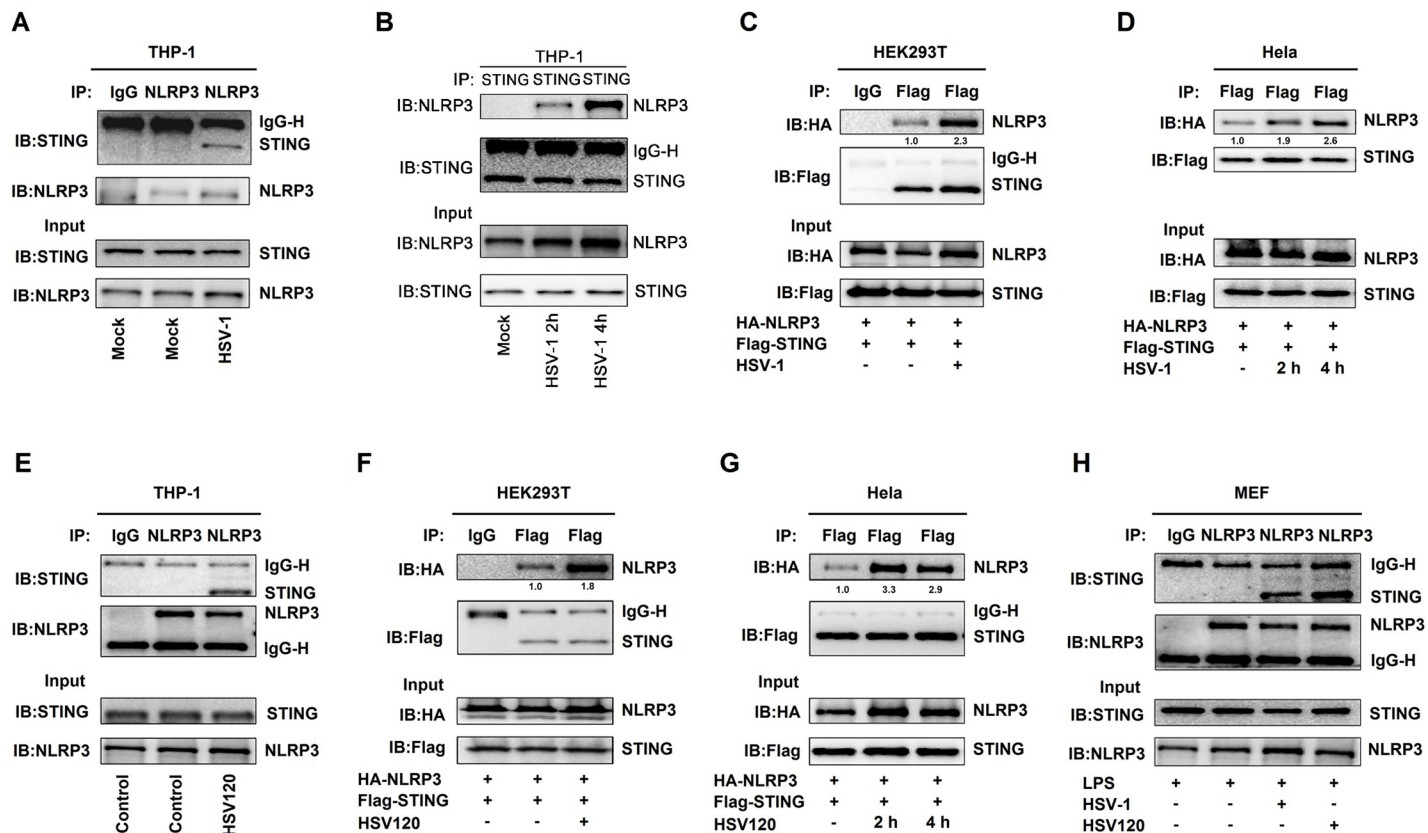

**Fig 2. HSV-1 infection promotes the STING-NLRP3 interaction.** (**A**, **B**) TPA-differentiated THP-1 macrophages were mock-infected or infected with HSV-1 (MOI = 1) for 4 h (A) or 2 h and 4 h (B). (**C**, **D**) HEK293T cells (C) and Hela cells (D) were co-transfected with pFlag-STING and pHA-NLRP3, and then infected with HSV-1 (MOI = 1) for 2–4 h. (**E**) TPA-differentiated THP-1 macrophages were transfected with HSV120 (3 μg/ml) by Lipo2000 for 4 h. The Lipo2000 was used as the control. (**F**, **G**) HEK293T cells (F) and Hela cells (G) were co-transfected with pFlag-STING and pHA-NLRP3 and then transfected with HSV120 (3 μg/ml) for 4 h. (**H**) Primary MEFs were primed with LPS (1 μg/ml) for 6 h, and then infected with HSV-1 (MOI = 1) for 4 h or transfected with HSV120 (3 μg/ml) for 4 h. (**A–H**) Cell lysates were subjected to Co-IP using IgG (Mouse) or anti-NLRP3 antibody (A), anti-STING antibody (B), IgG (Mouse) or anti-Flag antibody (C), anti-Flag antibody (D), IgG (Mouse) or anti-NLRP3 antibody (E), IgG (Mouse) or anti-Flag antibody (F), anti-Flag antibody (G), and IgG (Mouse) or anti-NLRP3 antibody (H), and then analyzed by immunoblotting using anti-NLRP3 and anti-STING antibody (top) or analyzed directly by immunoblotting using anti-NLRP3 and anti-STING antibody (as input) (bottom). Densitometry of the blots were measured by Image J.

in TPA-differentiated THP-1 macrophages, HEK293T cells, and HeLa cells, the NLRP3-STING interaction was promoted by HSV120 transfection (Fig 2E–2G). Moreover, in LPS-primed primary mouse embryo fibroblasts (MEFs), HSV-1 infection and HSV120 transfection facilitated endogenous STING-NLRP3 interaction (Fig 2H). Collectively, HSV-1 infection and HSV120 transfection facilitate the interaction of STING with NLRP3.

## HSV-1 infection induces IL-1β expression and secretion

IFI16 recognize HSV-1 DNA in the nucleus and then exits the nucleus and assembles with ASC and pro-caspase-1 to form an inflammasome complex in HFF cells. And HSV-1 infection can induce NLRP3-ASC interaction in HFF cells [24]. Next, we explored whether HSV-1 infection and HSV120 transfection regulate the inflammasome activation in THP-1 macrophages and LPS-primed mice primary MEFs. In TPA-differentiated THP-1 macrophages, endogenous IL-1β secretion was induced by Nigericin (a positive control for the inflammasome activation) [25] and HSV-1 (Fig 3A and 3B). Consistently, IL-1β maturation and Casp-1 cleavage, as well as pro-IL-1β production were activated upon HSV-1 infection (Fig 3C and

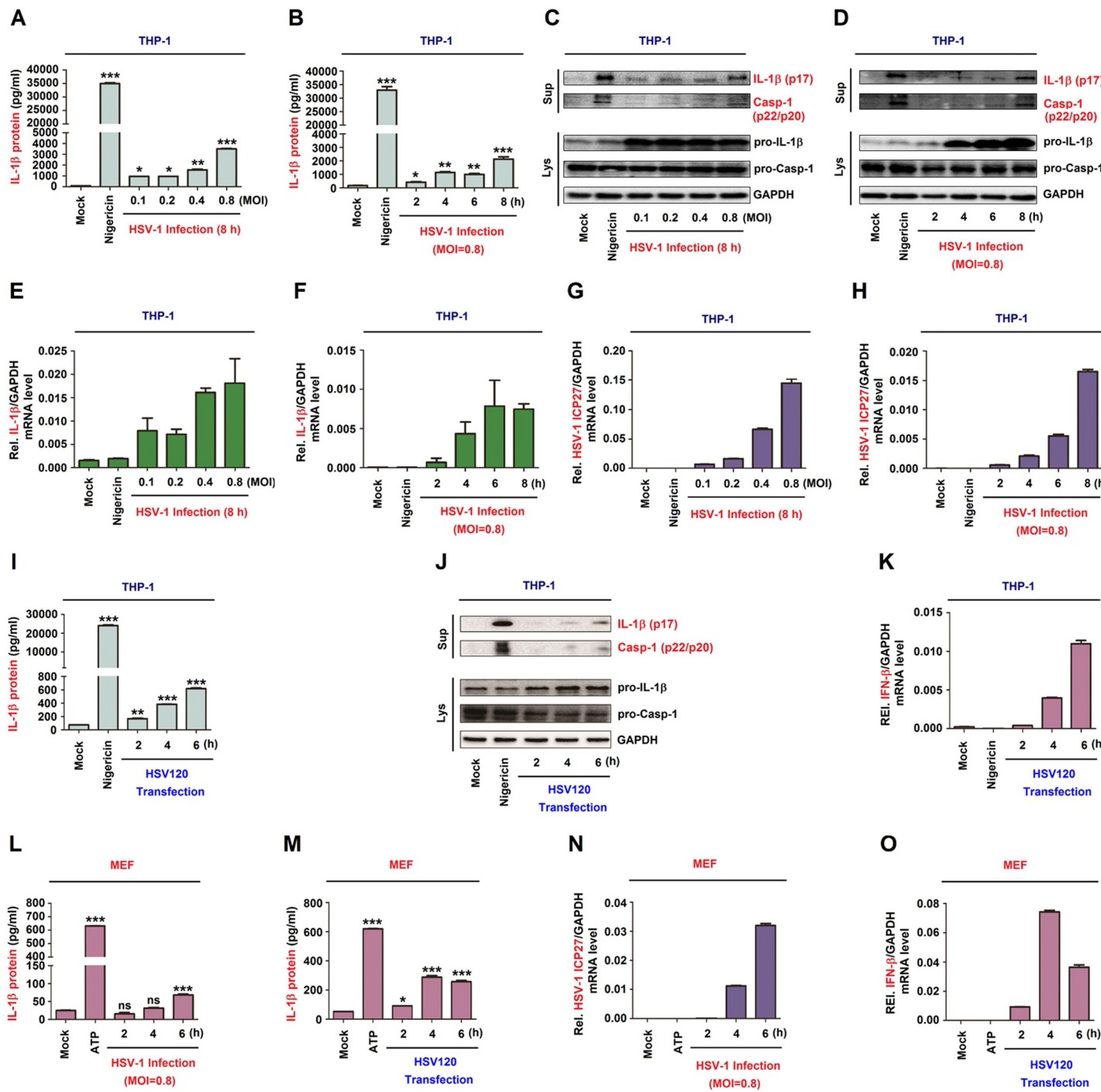

**Fig 3. HSV-1 infection induces IL-1β expression and secretion.** (A–K) TPA-differentiated THP-1 macrophages were treated with 2 μM Nigericin for 2 h, and infected with HSV-1 at MOI = 0.1, 0.2, 0.4 and 0.8 for 8 h (A, C, E and G), infected with HSV-1 at MOI = 0.8 for 2, 4, 6 and 8 h (B, D, F and H), or transfected with HSV120 (3 μg/ml) for 2, 4 and 6 h (I–K). (L–O) Primary MEFs were primed with LPS (1 μg/ml) for 6 h, and then treated with 5 mM ATP for 2 h or infected with HSV-1 (MOI = 1) (L and N) or transfected with HSV120 (3 μg/ml) (M, O) for 2, 4 and 6 h. (A, B, I, L and M) IL-1β protein was determined by ELISA. (C, D and J) Matured IL-1β (p17) and cleaved Casp-1 in supernatants (top) and pro-IL-1β production in lysates (bottom) were determined by Western-blot analyses. (E, F, G, H, K, N and O) IL-1β mRNA and GAPDH mRNA (E, F), HSV-1 ICP27 mRNA and GAPDH mRNA (G, H and N), and IFN-β mRNA and GAPDH mRNA (K, O) were quantified by RT-PCR. Data shown are means ± SEMs; *p < 0.05, **p < 0.01, ***p < 0.0001; ns, no significance.

3D). The expression of IFI16 was increased after HSV-1 infection (S1 Fig). Notably, IL-1β mRNA was not induced by Nigericin but induced upon HSV-1 infection (Fig 3E and 3F) and HSV-1 ICP27 mRNA was expressed in the infected cells (Fig 3G and 3H). Similarly, in TPA-differentiated THP-1 macrophages, IL-1β secretion was induced by Nigericin and facilitated by HSV120 (Fig 3I). IL-1β maturation and Casp-1 cleavage and pro-IL-1β production were stimulated by Nigericin and promoted by HSV120 (Fig 3J). IFN-β mRNA expression was not induced by Nigericin but activated by HSV120 (Fig 3K), demonstrating that HSV120 is effective in the cells. Moreover, in LPS-primed mice primary MEFs, endogenous IL-1β secretion was induced by ATP (a positive control), promoted upon HSV-1 infection, and enhanced by HSV120 stimulation (Fig 3L and 3M). HSV-1 ICP27 mRNA was detected in the cells (Fig 3N), suggesting that HSV-1 is replicated. IFN-β mRNA was not induced by ATP but activated by HSV120 in the cells (Fig 3O), demonstrating that HSV120 is effective. Therefore, IL-1β expression and secretion are induced upon HSV-1 infection and cytosolic DNA stimulation.

## The NLRP3 inflammasome is required for HSV-1-induced IL-1β activation

IFI16 and cGAS are innate DNA sensors mediating interferon-β-induction after HSV-1 infection [26, 27]. Here, we constructed THP-1 cell lines stably expressing negative control shRNA (sh-NC) and shRNAs (sh-IFI16 and sh-cGAS). HSV-1-induced, but not Nigericin-induced, IL-1β secretion (Fig 4A) as well as IL-1β (p17) cleavage and Casp-1 (p20 and p22) maturation were attenuated by sh-IFI16 and sh-cGAS (Fig 4B). The protein level of IFI16 and cGAS indicated that shRNAs are effective in the cells. HSV-1 ICP27 mRNA was expressed in infected cells (Fig 4C), confirming that HSV-1 is replicated in the cells. This result demonstrated that both IFI16 and cGAS are essential for HSV-1-induced NLRP3 inflammasome activation.

In macrophages, HSV-1 infection can induce inflammasome activation, but it is cytoplasmic DNA sensor AIM2-independent [28]. We constructed THP-1 cell lines stably expressing negative control shRNA (sh-NC) and shAIM2. IL-1β secretion (Fig 4D) as well as IL-1β (p17) cleavage and Casp-1 (p20 and p22) maturation (Fig 4E) induced by poly dA:dT, not activated by HSV-1 infection, were attenuated by sh-AIM2. The protein level of AIM2 indicated that shRNAs are effective in the cells. HSV-1 ICP27 mRNA was expressed in infected cells (Fig 4F), confirming that HSV-1 is replicated in the cells. Then, we determined whether the NLRP3 inflammasome is required for HSV-1 in the induction of IL-1β activation. Initially, the effects of glybenclamide (NLRP3 inhibitor) and Ac-YVAD-cmk (Casp-1 inhibitor) [29] on HSV-1-mediated IL-1β activation were evaluated. In TPA-differentiated THP-1 macrophages, IL-1β secretion (Fig 4G and 4H) as well as IL-1β (p17) cleavage and Casp-1(p20 and p22) maturation (Fig 4I and 4J) activated by Nigericin and HSV-1 were significantly attenuated by glybenclamide (Fig 4G and 4I) or Ac-YVAD-cmk (Fig 4H and 4J). HSV-1 ICP27 mRNA was expressed in infected cells (Fig 4K and 4L), indicating that HSV-1 is replicated.

In addition, the role of endogenous NLRP3 inflammasome in HSV-1-induced IL-1β activation was assessed in THP-1 cell lines stably expressing negative control shRNA (sh-NC) and shRNAs (sh-NLRP3, sh-ASC and sh-Casp-1) targeting the NLRP3 inflammasome components. Notably, IL-1β secretion (Fig 4M and 4O) as well as IL-1β (p17) cleavage and Casp-1 (p20 and p22) maturation (Fig 4N and 4P, top) induced by Nigericin (Fig 4M and 4N) or HSV-1 (Fig 4O and 4P) were attenuated by sh-NLRP3, sh-ASC, and sh-Casp-1. The NLRP3, ASC, and pro-Casp-1 proteins were own-regulated by sh-NLRP3, sh-ASC, and sh-Casp-1 stable cells, respectively (Fig 4N and 4P, bottom), indicating that shRNAs are effective in the cells. HSV-1 ICP27 mRNA was expressed in infected cells (Fig 4Q), confirming that HSV-1 is replicated in the cells.

Moreover, the direct role of NLRP3 in the regulation of HSV-1-induced IL-1β secretion was determined in LPS-primed primary MEFs of C57BL/6 WT pregnant mice and NLRP3[-/-]

pregnant mice. NLRP3 protein was detected in WT mice LPS-primed primary MEFs, but not in NLRP3[-/-] mice LPS-primed primary MEFs (Fig 4R), indicating that NLRP3 is knocked out in the null mice. IL-1β secretion was induced by ATP and HSV-1 in WT LPS-primed mice

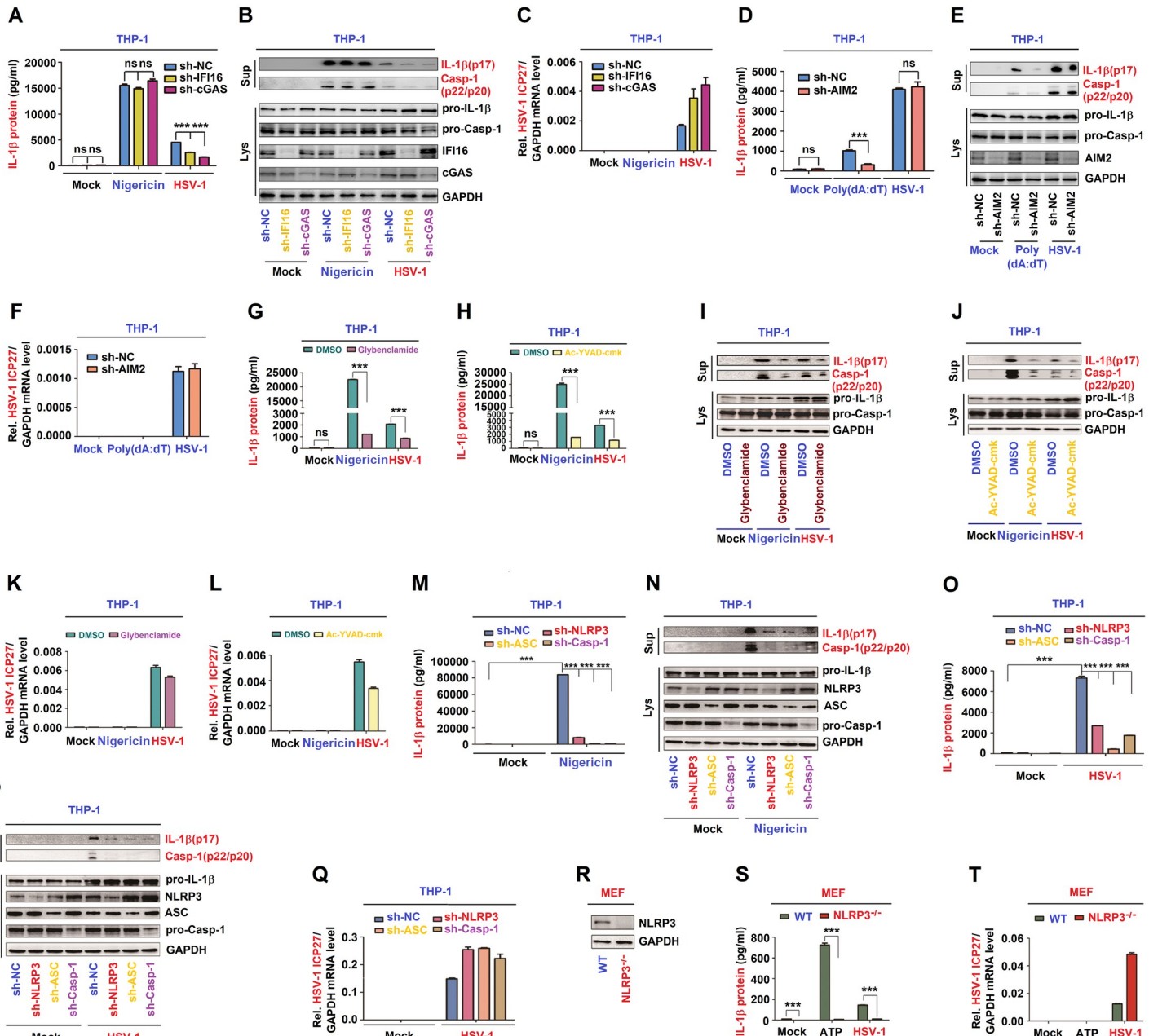

**Fig 4. The NLRP3 inflammasome is required for HSV-1-induced IL-1β activation.** (A–C) THP-1 cells stably expressing shRNAs targeting IFI16 or cGAS were generated and then treated with 2 µM Nigericin for 2 h or infected with HSV-1 at MOI = 0.8 for 8 h. (D–F) THP-1 cells stably expressing shRNAs targeting AIM2 were generated and then treated with 5 µg/ml poly(dA:dT) for 6 h or infected with HSV-1 at MOI = 0.8 for 8 h. (G–L) TPA-differentiated THP-1 macrophages were treated with 2 µM Nigericin for 2 h, infected with HSV-1 at MOI = 0.8 for 8 h, and then treated with Glybenclamide (25 µg/ml) (G, I and K) or Ac-YVAD-cmk (10 µg/ml) (H, J and L). (M–Q) THP-1 cells stably expressing shRNAs targeting NLRP3, ASC or Casp-1 were generated and then treated with 2 µM Nigericin for 2 h (M, N) or infected with HSV-1 at MOI = 0.8 for 8 h (O–Q). (R–T) Primary MEFs of C57BL/6 WT mice and C57BL/6 NLRP3[-/-] mice were primed with LPS (1 µg/ml) for 6 h, and then treated with 5 mM ATP for 2 h or infected with HSV-1 (MOI = 1) for 6 h. NLRP3 and GAPDH protein in lysates were determined by Western blot (R). IL-1β levels were determined by ELISA (A, D, G, I, M and S). Matured IL-1β (p17) and cleaved Casp-1 (p22/p20) in supernatants (top) or pro-IL-1β and pro-Casp-1 in lysates (bottom) were determined by Western-blot (B, E, H, N and P). HSV-1 ICP27 mRNA and GAPDH mRNA were quantified by RT-PCR (C, F, K, L, Q and T). Data shown are means ± SEMs; ***p < 0.0001; ns, no significance.

primary MEFs but not in NLRP3$^{-/-}$ mice LPS-primed primary MEFs (Fig 4S). HSV-1 ICP27 mRNA was expressed in infected cells, indicating that HSV-1 is replicated in the cells (Fig 4T). Collectively, inhibition, knock-down, and knock-out of the NLRP3 inflammasome components lead to the repression of IL-1β secretion and Casp-1 maturation, therefore the NLRP3 inflammasome is required for HSV-1-induced activation of IL-1β.

## STING recruits NLRP3 to the ER to promote the inflammasome formation

STING predominantly resides in the endoplasmic reticulum (ER) to regulate innate immune signaling processes [18]. Here we evaluated whether STING promotes NLRP3 ER localization and activates the inflammasome. In HeLa cells, NLRP3 alone diffusely distributed in the cytoplasm and STING or STING(delTM5) alone located in the ER (as indicated by the ER marker, ER blue), while NLRP3 and STING, but not STING(delTM5), together co-localized and distributed in the ER to form specks (Fig 5A), which is an indication of the NLRP3 inflammasome complex formation [22], suggesting that STING facilitates NLRP3 ER localization and promotes the inflammasome formation. In addition, NLRP3 diffusely distributed in the cytosol of untreated cells, but formed distinct specks upon HSV-1 infection or HSV120 transfection (Fig 5B). Moreover, NLRP3 diffusely distributed in the cytosol of untreated cells, while upon HSV-1 infection or HSV120 transfection, NLRP3 forms distinct specks in the ER as indicated by Calnexin (ER protein) (Fig 5C) and ER blue (ER marker) (Fig 5D), but not in TGN (S2A Fig) and Cis Golgi (S2B Fig). In THP-1 macrophages, we also found the improvement of NLRP3 localization in the ER after the HSV-1 infection (Fig 5E), but not in TGN (S2C Fig). Notably, in transfected HeLa cells, NLRP3, STING, and Calnexin were detected in whole cell lysate (WCL) and purified ER fraction, and interestingly, NLRP3 abundance was enhanced by STING in purified ER fraction (Fig 5F). Similarly, in mock-infected THP-1 macrophages and LPS-primed mice primary MEFs, NLRP3, STING, and Calnexin were detected in WCL and purified ER, and NLRP3 abundance was increased in the ER upon HSV-1 infection (Fig 5G and 5H). Therefore, STING, HSV-1, and HSV120 facilitate the NLRP3 inflammasome formation in the ER.

Moreover, the effect of endogenous STING on NLRP3 translocation to the ER was further determined by using shRNA targeting STING (sh-STING). Hela cells stably expressing sh-NC or sh-STING were generated, and then transfected with Flag-NLRP3 and infected with HSV-1 or transfected with HSV120. In the absence of sh-STING, NLRP3 diffusely distributed in the cytosol of untreated cells and formed distinct specks in the ER, as indicated by Calnexin and ER Blue (Fig 5I and 5J, top), upon HSV-1 infection or HSV120-transfection, however, in the presence of sh-STING, NLRP3 failed to form specks upon HSV-1 infection or HSV120 transfection (Fig 5I and 5J, bottom), indicating that STING knock-down leads to the repression of HSV-1-induced formation of the NLRP3 inflammasome. In addition, NLRP3, Calnexin, and STING were detected in WCL and purified ER fraction of Hela cells (Fig 5K), THP-1 cells (Fig 5L) and LPS-primed mice primary MEFs (Fig 5M), and notably, NLRP3 level was higher in purified ER fraction upon HSV-1 infection in sh-NC stable cells (Fig 5K–5M, lane 6 vs. 5) as compared with sh-STING stable cells (Fig 5K–5M, lane 8 vs. 7), suggesting that STING knock-down results in the attenuation of NLRP3 localization in the ER upon HSV-1 infection. We also confirmed that STING abundance was down-regulated by sh-STING (Fig 5K–5M). Collectively, STING improves NLRP3 ER localization and promotes the NLRP3 inflammasome formation upon HSV-1 infection and cytosolic DNA stimulation.

## STING deubiquitinates NLRP3 to activate the NLRP3 inflammasome

The deubiquitination of NLRP3 is required for the NLRP3 inflammasome activation [30]. We next investigated whether STING plays a role in the deubiquitination of NLRP3, thereby

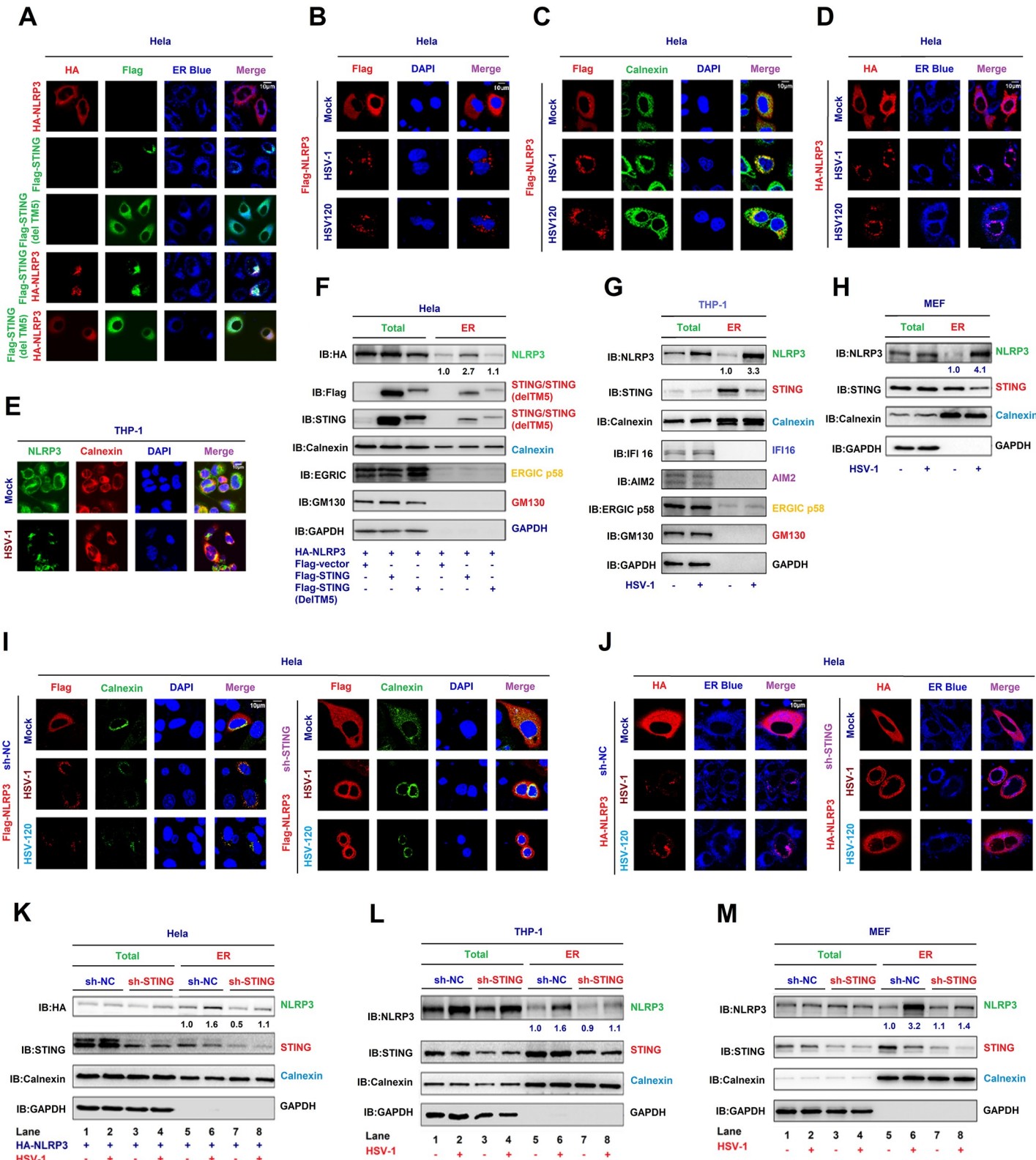

**Fig 5. STING recruits NLRP3 to the ER and promotes the inflammasome formation.** (**A–D**) Hela cells were co-transfected with pFlag-STING, TM5-deleted STING (delTM5) and pHA-NLRP3 (A), transfected with pFlag-NLRP3 and infected with HSV-1 (MOI = 1) for 4 h or transfected with HSV120 (3 μg/ml) for 4 h (B, C and D). Sub-cellular localization of HA-NLRP3 (green) (A and D), Flag-STING (red) or TM5-deleted STING(delTM5) (green) (A), ER Blue (blue) (A, D), Flag-NLRP3 (red) (B,

C), Calnexin (green) (B), DAPI (blue) (B and C), and Calnexin (green) (C) were examined by confocal microscopy. (**E**) TPA-differentiated THP-1 macrophages were infected with mock or HSV-1 (MOI = 1) for 4 h. Sub-cellular localization of NLRP3 (green), Calnexin (red) and DAPI (blue) were examined by confocal microscopy. (**F–H**) Hela cells were co-transfected with pFlag-STING and pHA-NLRP3 (F). THP-1 macrophages were infected with HSV-1 (MOI = 1) for 4 h (G). Primary MEFs were primed with LPS (1 μg/ml) for 6 h and infected with HSV-1 (MOI = 1) for 4 h (H). HA-NLRP3, Flag-STING, STING, ERGIC p58, GM130 (cis Golgi), Calnexin (ER), IFI16, AIM2 and GAPDH in whole cell lysate (WCL) and purified ER were determined by Western-blot analyses. (**I, J**) Hela cells stably expressing sh-STNG were transfected with pFlag-NLRP3 and infected with HSV-1 (MOI = 1) for 4 h or transfected with HSV120 (3 μg/ml) for 4 h. Sub-cellular localization of Flag-NLRP3 (red) (I), Calnexin (green) (I), DAPI (blue) (I), HA-NLRP3 (red) (J) and ER blue (J) were examined by confocal microscopy. (**K–M**) Hela cells stably expressing sh-STING were transfected with pHA-NLRP3 and infected with HSV-1 (MOI = 1) for 4 h (K). THP-1 cells stably expressing sh-STING were differentiated to macrophages, and infected with HSV-1 (MOI = 1) for 4 h (L). Primary MEFs stably expressing sh-STING were primed with LPS (1 μg/ml) for 6 h, and infected with HSV-1 (MOI = 1) for 4 h (M). HA-NLRP3 (K), NLRP3 (L, M), STING (K–M), Calnexin (ER) (K–M) and GAPDH (K–M) in WCL and purified ER fraction were determined by Western-blot analyses. Densitometry of the blots were measured by Image J.

facilitating the inflammasome activation. Interestingly, NLRP3 polyubiquitination catalyzed by HA-UB, HA-UB(K48R), or HA-Ub(K63R) was repressed by STING (Fig 6A and 6B). We also revealed that NLRP3 polyubiquitination catalyzed by HA-UB, HA-UB(K48O) (ubiquitin mutant that only retains a single lysine residue), or HA-UB(K63O) (ubiquitin mutant that only retains a single lysine residue) was suppressed by STING (Fig 6C). These results reveal that STING decreases K48- and K63-linked polyubiquitination of NLRP3.

In addition, we examined whether HSV-1 infection and HSV120 transfection regulate the deubiquitination of NLRP3. Notably, NLRP3 polyubiquitination catalyzed by HA-UB was attenuated upon HSV-1 infection (Fig 6D) or by HSV120 transfection (Fig 6E). In THP-1 differentiated macrophages, UB-catalyzed (Fig 6F, top), K48-linked (Fig 6F, middle), and K63-linked (Fig 6F, bottom) polyubiquitination of endogenous NLRP3 was repressed upon HSV-1 infection; and UB-catalyzed polyubiquitinion of endogenous NLRP3 was repressed by HSV120 (Fig 6G). Moreover, in LPS-primed mice primary MEFs, UB-catalyzed ployubiquitination of endogenous NLRP3 was attenuated upon HSV-1 infection (Fig 6H). Therefore, HSV-1 infection and HSV120 stimulation promote the deubiquitination of endogenous NLRP3. Moreover, upon HSV-1 infection, the ployubiquitination of endogenous NLRP3 was attenuated in the presence of sh-NC but relatively unaffected in the presence of sh-STING (Fig 6I), indicating that STING knock-down leads to repression of NLRP3 deubiquitination. Taken together, STING attenuates K48- and K63-linked polyubiquitination of NLRP3 to promote the inflammasome activation upon HSV-1 infection and cytosolic DNA stimulation.

## STING is required for the NLRP3 inflammasome activation upon DNA virus infection

Since STING recruits NLRP3 to the ER and removes NLRP3 deubiquitination upon HSV-1 infection, we speculated that STING may play a role in HSV-1-induced NLRP3 inflammasome activation. The effect of STING knock-down on HSV-1-induced NLRP3 inflammasome activation was initially examined in THP-1 cells stably expressing sh-STING. Endogenous IL-1β secretion as well as IL-1β (p17) cleavage and Casp-1 (p20 and p22) maturation induced by HSV-1 were significantly attenuated by sh-STING (Fig 7A and 7B). HSV-1 ICP27 mRNA was detected in HSV-1 infection cells (Fig 7C), indicating that HSV-1 is replicated. Additionally, endogenous IL-1β secretion, IL-1β (p17) maturation, and Casp-1 (p20 and p22) cleavage induced by DNA90, HSV120, or HSV-1 were suppressed by sh-STING (Fig 7D and 7E). Thus, STING knock-down leads to the suppression of IL-1β secretion and Casp-1 maturation upon DNA virus infection and cytosolic DNA stimulation.

Accordingly, we determine whether STING plays a specific role in the NLRP3 inflammasome activation mediated by DNA virus. THP-1 cells stably expressing sh-STING were differentiated to macrophages, and then treated with Nigericin or infected with HSV-1. Notably, sh-STING significantly attenuated endogenous IL-1β secretion as well as L-1β (p17) maturation

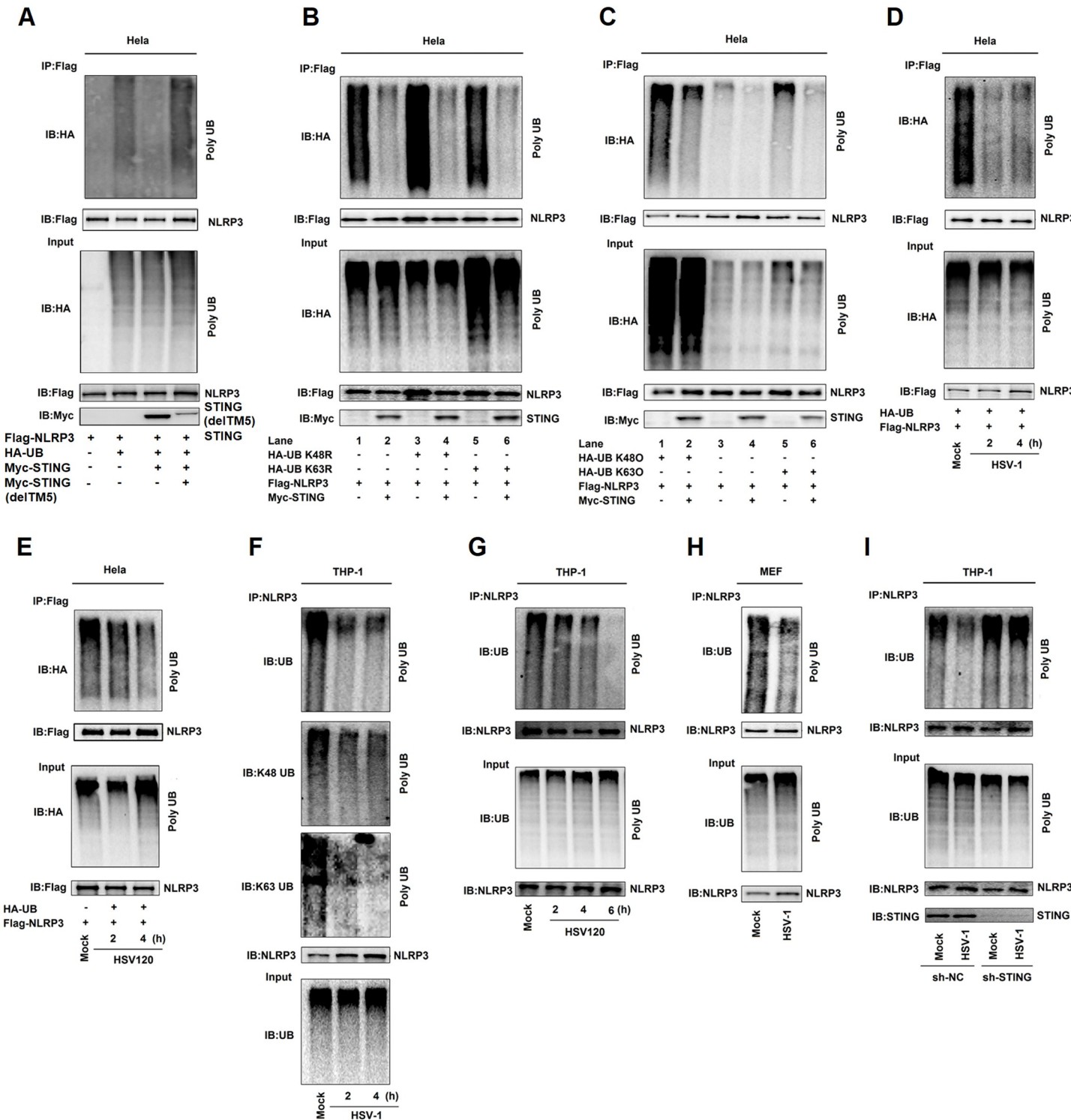

**Fig 6. STING deubiquitinates NLRP3 to activate the NLRP3 inflammasome.** (A–E) Hela cells were co-transfected with pFlag-NLRP3, pHA-Ubiquitin, and pMyc-STING (A), with pFlag-NLRP3, pHA-Ubiquitin, pHA-Ubiquitin mutations (K48R), pHA-Ubiquitin mutations (K63R) and pMyc-STING (B), with pFlag-NLRP3, pHA-Ubiquitin, pHA-Ubiquitin mutations(K48O), pHA-Ubiquitin mutations(K63O), and pMyc-STING (C), with pFlag-NLRP3 and pHA-Ubiquitin and then infected with HSV-1(MOI = 1) for 2 and 4 h (D), and transfected with HSV120 (3 µg/ml) for 2 and 4 h (E). Cell lysates were prepared and subjected to denature-IP using anti-Flag antibody and then analyzed by immunoblotting using an anti-HA or anti-Flag antibody (top) or subjected directly to Western blot using an anti-Flag, anti-HA or anti-Myc antibody (as input) (bottom). (**F–I**) TPA-differentiated THP-1 macrophages were infected with HSV-1 (MOI = 1) for 2 and 4 h (F), and transfected with HSV120 (3 µg/ml) for 2, 4 and 6 h (G). Mice primary MEFs were infected with HSV-1 (MOI = 1) for 2 h (H). THP-1 cells stably expressing sh-NC or sh-STING were generated and

differentiated to macrophages and then infected with HSV-1 (MOI = 1) for 4 h (I). Cell lysates were prepared and subjected to denature-IP using anti-NLRP3 antibody and then analyzed by immunoblotting using an anti-Ubiquitin, anti-K48-Ubiquitin, anti-K63-Ubiquitin or anti-NLRP3 antibody (top) (F) or using anti-NLRP3 antibody (G, H and I), or subjected directly to Western blot using an anti- Ubiquitin or anti-NLRP3 antibody (as input) (bottom).

and Casp-1 (p20 and p22) cleavage, or ASC oligomerization induced upon HSV-1 infection, but had no effect on IL-1β secretion or IL-1β (p17) maturation and Casp-1 (p20 and p22) cleavage, or ASC oligomerization induced by Nigericin stimulation (Fig 7F–7I). HSV-1 ICP27 mRNA was detected in HSV-1 infection cells (Fig 7H), indicating that HSV-1 is replicated in the cells. Moreover, we explored whether STING plays roles in the NLRP3 inflammasome activation induced by RNA viruses. THP-1 differentiated macrophages stably expressing sh-STING were infected with RNA viruses, Sendai virus (SeV) and Zika virus (ZIKV), or with HSV-1. Interestingly, endogenous IL-1β secretion as well as IL-1β (p17) maturation and Casp-1 (p20 and p22) cleavage were induced upon the infections of the three viruses, however, HSV-1-mediated induction was attenuated by sh-STING, but SeV- or ZIKV-mediated inductions were not affected by sh-STING (Fig 7J and 7K). SeV P mRNA, ZIKV mRNA, and HSV-1

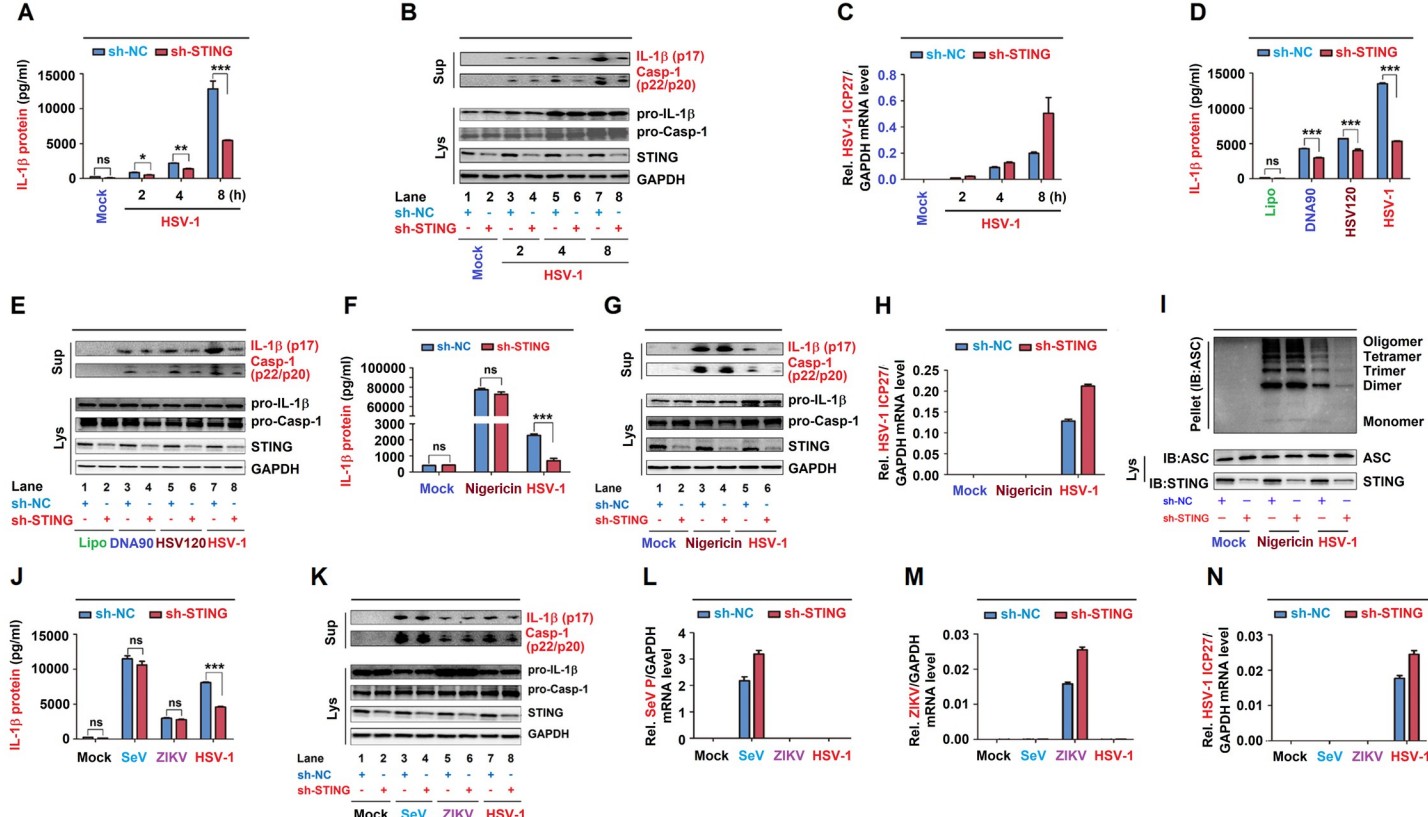

**Fig 7. STING is required for the NLRP3 inflammasome activation upon DNA virus infection.** (A–I) THP-1 cells stably expressing sh-NC or sh-STING were generated and differentiated to macrophages, and then infected with HSV-1 at MOI = 0.8 for 2, 4 and 8 h (A–C), transfected with DNA90 (3 μg/ml), HSV120 (3 μg/ml) or infected with HSV-1 at MOI = 0.8 for 8 h (D, E), and treated with 2 μM Nigericin for 2 h and infected with HSV-1 at MOI = 0.8 for 8 h (F–I). IL-1β levels were determined by ELISA (A, D and F). Matured IL-1β (p17) and cleaved Casp-1 (p22/p20) in supernatants or STING, pro-IL-1β and pro-Casp-1 in lysates were determined by Western-blot (B, C and G). HSV-1 ICP27 mRNA and GAPDH mRNA were quantified by RT-PCR (C, H). Pellets were subjected to ASC oligomerization assays (top) and lysates were prepared for Western blots (bottom) (I). (J–N) THP-1 cells stably expressing shRNA targeting STING was generated and differentiated to macrophages, then infected with SeV (MOI = 1) for 24 h, ZIKV (MOI = 1) for 24 h or HSV-1 (MOI = 0.8) for 8 h. IL-1β levels were determined by ELISA (J). Matured IL-1β (p17) and cleaved Casp-1 (p22/p20) in supernatants or STING, pro-IL-1β and pro-Casp-1 in lysates were determined by Western-blot (K). SeV P mRNA (L), ZIKV mRNA (M), HSV-1 ICP27 mRNA (N) and GAPDH mRNA were quantified by RT-PCR. Data shown are means ± SEMs; *p < 0.05, **p < 0.01, ***p < 0.0001; ns, no significance.

ICP27 mRNA were detected in infected cells, respectively (Fig 7L–7N). Taken together, STING plays specific roles in the NLRP3 inflammasome activation upon DNA virus infection or cytosolic DNA stimulation, but has no effect on the inflammasome activation induced by RNA virus infection or Nigericin stimulation.

## NLRP3 is critical for host defense against HSV-1 infection in mice

To gain insights into the biological function of NLRP3 *in vivo*, we analyzed C57BL/6 NLRP3$^{+/+}$ WT mice and C57BL/6 NLRP3$^{-/-}$ deficiency mice. Notably, HSV-1-infected NLRP3$^{-/-}$ mice began to die at 5 days post-infection and all infected mice died at 7 days post-infection, while infected WT mice began to die at 7 days post-infection and 30% WT mice was survival after 11 days post-infection (Fig 8A). The body weights of infected NLRP3$^{-/-}$ mice decreased continuously until died, while the body weights of infected WT mice gradually decreased until 7 days post-infection and then gradually increased (Fig 8B). Thus, NLRP3 deficiency mice are more susceptibility to HSV-1 infection and exhibit early onset of death upon the infection.

Notably, in the mice blood, IL-1β secretion was induced upon HSV-1 infection in WT mice, whereas it was not induced in NLRP3$^{-/-}$ mice (Fig 8C), IL-1β mRNA level was higher in WT mice as compared with NLRP3$^{-/-}$ mice (Fig 8D), IL-6 mRNA and TNF-α mRNA were no significant difference between WT and NLRP3$^{-/-}$ mice (Fig 8E and 8F), and HSV-1 UL30 mRNA was expressed in infected WT and NLRP3$^{-/-}$ mice (Fig 8G). These results indicate that NLRP3 deficiency leads to the repression of IL-1β expression and secretion in mice. Interestingly, in HSV-1 infected mice lung and brain, IL-1β mRNA and IL-6 mRNA were significantly higher in WT mice as compared with NLRP3$^{-/-}$ mice (Fig 8H, 8I, 8K and 8L), however, the viral titers were much lower in WT mice as compared with NLRP3$^{-/-}$ mice (Fig 8J and 8M), suggesting that NLRP3 deficiency results in the attenuation of IL-1β expression and the promotion of HSV-1 replication in mice lung and brain. Moreover, Hematoxylin and Eosin (H&E) staining showed that more infiltrated neutrophils and mononuclear cells were detected in the lung (Fig 8N, left) and brain (Fig 8O, left) of mock infected WT mice as compared with HSV-1 infected WT mice, and this increase was eliminated in HSV-1 infected WT mice. Immunohistochemistry (IHC) analysis revealed that IL-1β protein level was higher in the lung (Fig 8N, right) and brain (Fig 8O, right) of mock infected WT mice as compared with HSV-1 infected WT mice, and this increase was eliminated in HSV-1 infected HSV-1 infected WT mice, revealing that NLRP3 deficiency mice are more susceptibility to HSV-1 infection and elicit weak inflammatory responses. Collectively, we propose that NLRP3 is essential for host defense against HSV-1 infection by facilitating IL-1β activation (Fig 9).

## Discussion

This study reveals a distinct mechanism by which the cGAS-STING-NLRP3 pathway promotes the NLRP3 inflammasome activation and IL-1β secretion upon DNA virus infection and cytosolic DNA stimulation. The cGAS-STING pathway mediates immune defense against infection of DNA-containing pathogens and detects tumor-derived DNA and generates intrinsic antitumor immunity [31]. More recent studies reported that in human monocytes, the cGAS-STING pathway is essential for cytosolic DNA induced-NLRP3 inflammasome [12] and in mice BMDMs, the cGAS-STING pathway is required for Chlamydia trachomatis-induced inflammasome activation and IL-1β secretion [13, 14]. Our results are consistent with the reports and further support that the cGAS-STING pathway is essential not only for cytosolic DNA induced- or Chlamydia trachomatis-induced NLRP3 inflammasome activation, but also for DNA virus-induced NLRP3 inflammasome activation. This study also further reveals that the cGAS-STING pathway is required for the NLRP3 inflammasome activation not only in

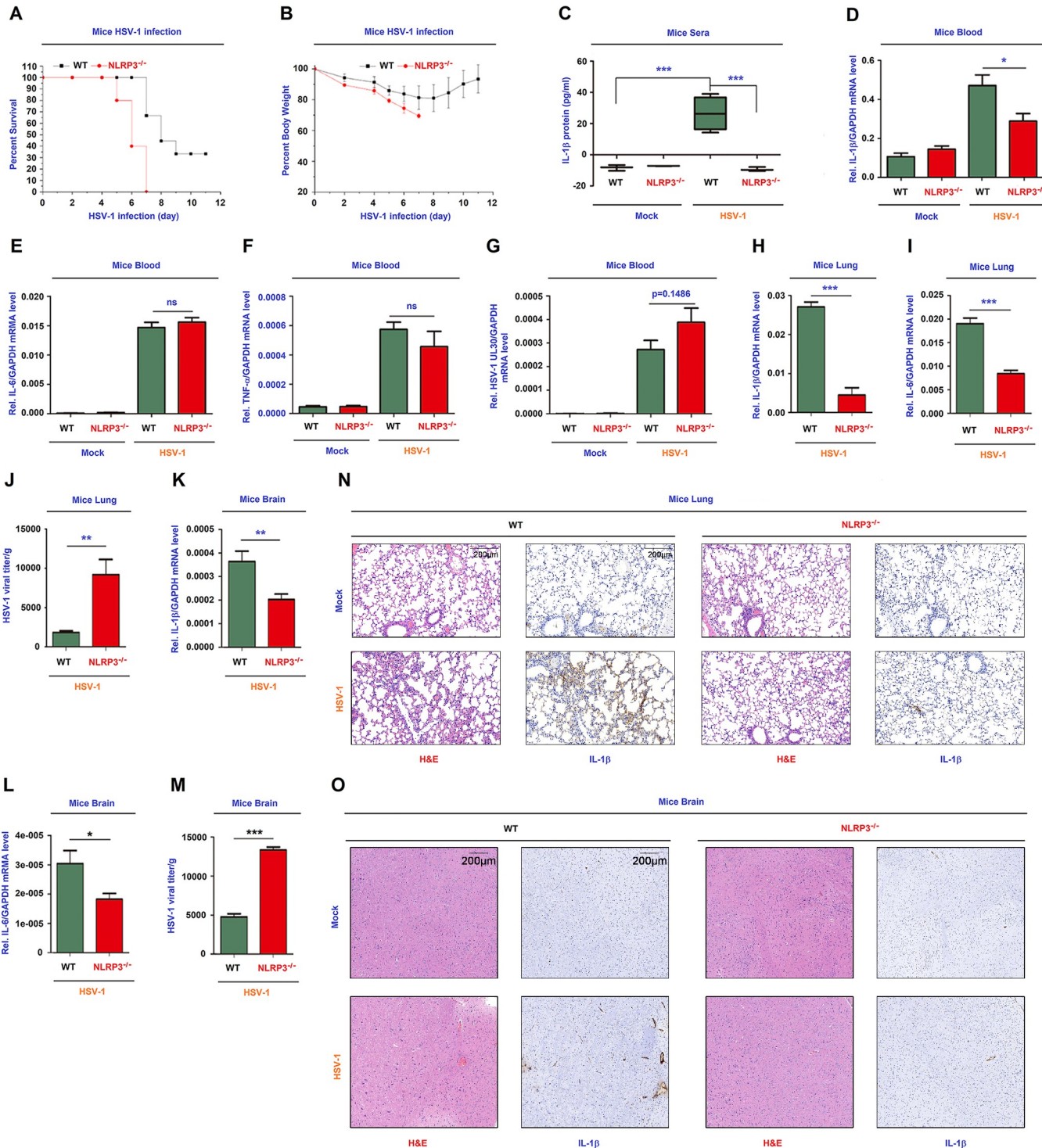

**Fig 8. NLRP3 is critical for host defense against HSV-1 infection in mice. (A, B)** C57BL/6 WT mice (8-week-old, female, n = 9) and C57BL/6 NLRP3[-/-] mice (8-week-old, female, n = 5) were infected i.p. with HSV-1 at $1\times10^7$ pfu for 11 days. The survival rates (A) and body weights (B) of mice were evaluated. **(C–G)** WT mice (8-week-old, female) and NLRP3[-/-] mice (8-week-old, female) were mock-infected i.p. with PBS (WT and NLRP3[-/-] mice, n = 3) or infected i.p. with HSV-1 (WT mice, n = 4, NLRP3[-/-] mice, n = 3) at $1\times10^7$ pfu for 6 h. IL-1β in mice sera was determined by ELISA (C). IL-1β mRNA (D), IL-6 mRNA (E), TNF-α mRNA (F), HSV-1 UL30 mRNA (G) and GAPDH mRNA in mice blood were quantified by RT-PCR. **(H–J)** WT mice (8-week-old, female, n = 4) and NLRP3[-/-] mice (8-week-old, female, n = 4) were infected i.p. with HSV-1 at $1\times10^7$ pfu for 1 days. IL-1β mRNA (H), IL-6 mRNA (I) and GAPDH mRNA in mice lung were quantified by RT-PCR. HSV-1 viral titers were measured by plaque assays for mice lung (J). **(K–M)** WT mice (8-week-old, female, n = 7) and NLRP3[-/-] mice (8-week-old, female, n = 7) were infected i.p. with HSV-1 at $1\times10^7$ pfu for 4 days. IL-1β mRNA (K), IL-6 mRNA (L) and GAPDH mRNA in mice brain were

quantified by RT-PCR. HSV-1 viral titers were measured by plaque assays for mice brain (M). (**N**) WT mice (8-week-old, female) and NLRP3[-/-] mice (8-week-old, female) were infected i.p. with HSV-1 at $1\times10^7$ pfu for 1 day. The lung tissues were stained with histological analysis (H&E or IL-1β). (**O**) WT mice (8-week-old, female) and NLRP3[-/-] mice (8-week-old, female) were infected i.p. with HSV-1 at $1\times10^7$ pfu for 4 days. The brain tissues were stained with histological analysis (H&E or IL-1β). Data shown are means ± SEMs; *p < 0.05, **p < 0.01, ***p < 0.0001; ns, no significance.

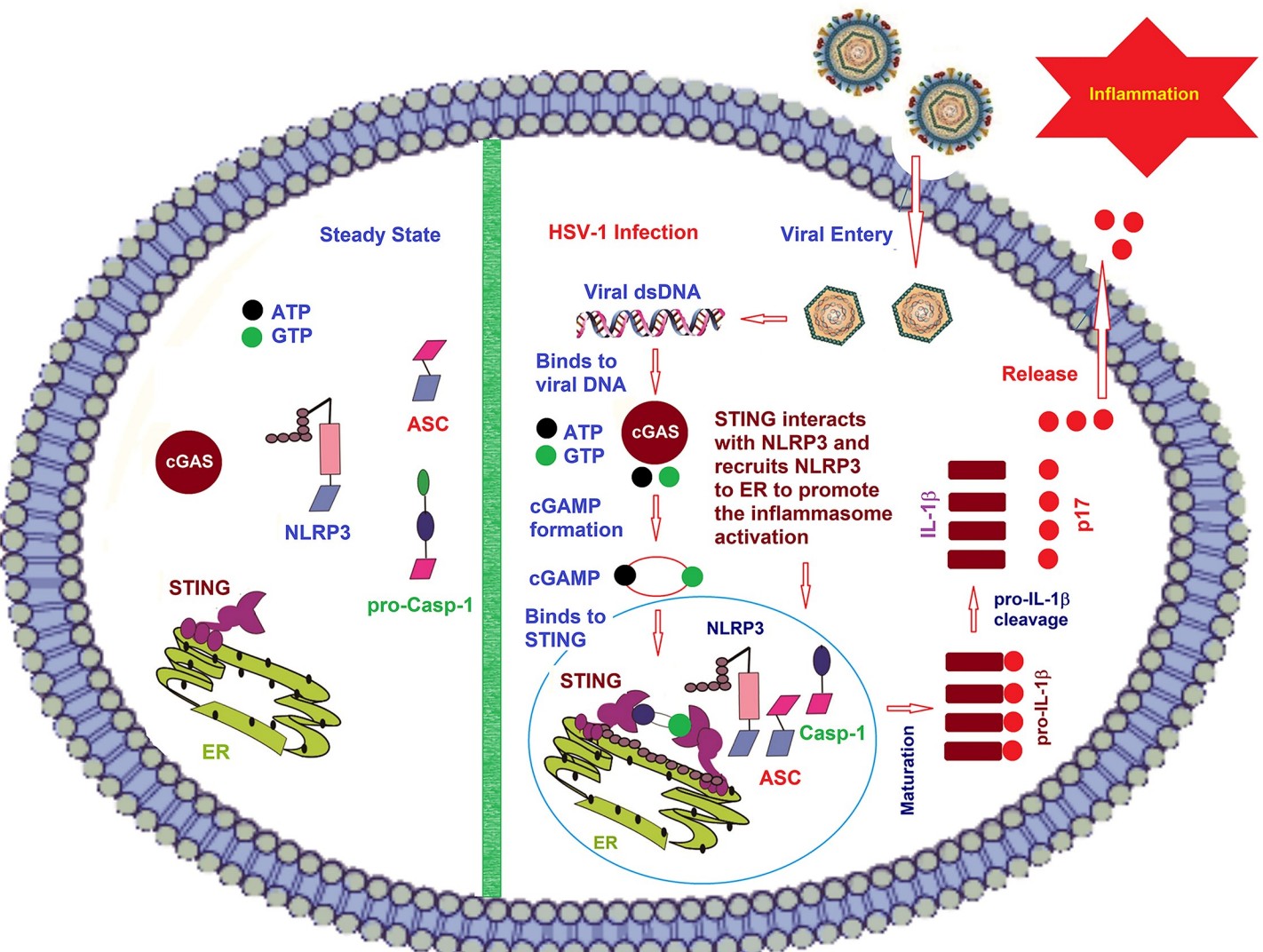

**Fig 9. A proposed model for the regulation of NLRP3 inflammasome activation mediated by the cGAS-STING-NLRP3 pathway.** Left: In the mock infection, all these proteins are in the cytoplasm. NLRP3 contains an N-terminal pyrin domain (PYD), a NACHT-associated domain (NAD) and 7 C-terminal leucine rich repeats (LRR). ASC (the apoptosis-associated speck-like protein with CARD domain) has two domains (PYD and CARD). The effecter protein pro-Casp-1 has the three domains (CARD, p20 and p10). It has been demonstrated that the NLR protein NLRP3, together with the adaptor protein, apoptosis-associated speck-like protein with CARD domain (ASC), promotes the cleavage of the pro-Casp-1 to generate active subunits p20 and p10, which regulate the maturation of IL-1β.STING comprises 5 putative transmembrane (TM) regions, predominantly resides in the endoplasmic reticulum (ER) and regulates innate immune signaling processes. Right: After the infection by HSV-1, everything has changed. Herpes simplex virus type 1 (HSV-1) belongs to the family of Herpesviridae, which causes various mild clinical symptoms. HSV-1 genomic DNA then induce the cGAS-cGAMP-STING pathway for the IFN-I production. Upon binding of cytoplasmic viral DNA, GAS (cyclic GMP-AMP synthase) catalyzed the formation of the second messenger cGAMP, which is an endogenous second messenger by binding to STING and inducing STING's dimerization. After that, HSV-1 infection also induced NLRP3 translocation to Endoplasmic Reticulum (ER) by STING through the interaction between STING and NLRP3. Then STING promotes the deubiquitination of NLRP3 in K48-linked and K63-linked polyubiquitin chains and initiates the activation of NLRP3 inflammasome, promotes the cleavage of the pro-Casp-1 to generate active subunits p20 and p10, which regulate the maturation of IL-1β, and IL-1β then secreted outside the cells.

human monocytes and mice BMDMs, but also in human embryonic kidney cells (HEK293T), Hela cells, human leukemic monocytes/macrophages (THP-1), and mice primary mouse embryo fibroblasts (MEFs). More interestingly, our results reveal a distinct mechanism underlying STING-mediated NLRP3 inflammasome activation, and demonstrate for the first time that STING binds to NLRP3 and promotes the inflammasome activation through two approaches. First, STING binds to and improves NLRP3 localization in ER to promote the formation of the NLRP3 inflammasome. Second, STING interacts with NLRP3 and attenuates K48- and K63-linked polyubiquitination of NLRP3 to induce the activation of the NLRP3 inflammasome. Notably, upon HSV-1 infection and HSV120 stimulation, STING binds to NLRP3, promotes the NLRP3-ASC interaction (an indicator of inflammasome complex assembly) [20], facilitates NLRP3-mediated ASC oligomerization (a critical step for inflammasome activation) [21], enhances NLRP3 to form specks (an indicator of inflammasome activation) [22], and enhances IL-1β secretion (a fundamental reaction of the inflammatory responses) [7]. Collectively, the cGAS-STING-NLRP3 pathway plays key roles in the NLRP3 inflammasome activation and IL-1β secretion upon DNA virus infection and cytosolic DNA stimulation.

Notably, STING knock-down attenuates the NLRP3 inflammasome activation mediated upon DNA virus infection or cytosolic DNA stimulation, but has no effect on the NLRP3 inflammasome activation induced by RNA virus infection or Nigericin induction. Many RNA viruses induce the NLRP3 inflammasome activation, including influenza A virus (IAV) [32], Vesicular mastitis virus (VSV) and Encephalomyocarditis virus (EMCV) [33], Measles virus (MV) [34], West Nile virus (WNV) [35], Rabies virus (RV) [36], Hepatitis C virus (HCV) [37], Dengue virus (DENV) [38], Enterovirus 71 (EV71) [19] and Zika virus (ZIKV) [17]. Some DNA viruses also regulate the NLRP3 inflammasome activation, such as Adenovirus (AdV) [39] and HSV-1 [24]. HSV-1 VP22 inhibits AIM2-denpendent inflammasome activation so that HSV-1 infection of macrophages-induced inflammasome activation is AIM2-independent [28]. Here, we demonstrate that the cGAS-STING-NLRP3 pathway is required for HSV-1-induced NLRP3 inflammasome activation and critical for host defense against DNA virus infection.

The mechanisms of NLRP3 inflammasome activation have been intensely studied. The mitochondria-associated adaptor protein (MAVS) promotes NLRP3 mitochondrial localization and the inflammasome activation [40]. PtdIns4P mediates the NLRP3 inflammasome activation in trans-Golgi network (TGN) [41]. NLRP3 associates with SCAP-SREBP2 to form a ternary complex that translocates to the Golgi apparatus for optimal inflammasome assembly [42]. Here we find that STING promotes NLRP3 translocation to the ER and facilitates the inflammasome activation. Moreover, post-translational modifications of NLRP3 are critical for its activation, including phosphorylation [43], SUMOylation [44], and ubiquitination [45]. MARCH7 and TRIM31 facilitate NLRP3 ubiquitination and proteasomal degradation [46, 47]. Pellino2 promotes K63-linked ubiquitination of NLRP3 as part of the priming phase [48]. Interestingly, we demonstrate that STING removes K48- and K63-linked ubiquitination of NLRP3 to promote the inflammasome activation, and reveal that HSV-1 infection induces STING-mediated deubiquitination of NLRP3.

Moreover, NLRP3 is related to many human diseases. Fibrillar amyloid-β peptide, the major component of Alzheimer's disease brain plaques, facilitates the NLRP3 inflammasome activation [49]. Monosodium urate (MSU) crystals induce the autoinflammatory disease gout and activate the NLRP3 inflammasome [50]. NLRP3, IL-1β, reactive oxygen species (ROS), and TXNIP are implicated in the type 2 diabetes mellitus (T2DM) pathogenesis [51]. Our study gains insights into the biological function of the cGAS-STING-NLRP3 pathway in host defense against HSV-1 infection in mice. NLRP3 deficiency mice are more susceptibility to

HSV-1 infection, exhibit early onset of death upon infection, represses IL-1β secretion, and elicits robust inflammatory responses in the tissues. Collectively, these results demonstrate that NLRP3 is essential for host defense against HSV-1 infection by inducting IL-1β expression and secretion.

In conclusion, we reveal a distinct mechanism underlying the regulation of the NLRP3 inflammasome activation upon HSV-1 infection. In this model, STING (the central molecule of the antiviral and inflammatory immune pathways) interacts with NLRP3 (the key component of the inflammasomes), decreases NLRP3 polyubiquitination, improves the localization of NLRP3 in ER, and facilitates the NLRP3 inflammasome activation, thereby inducing IL-1β secretion upon DNA virus infection and cytosolic DNA stimulation.

## Materials and methods

### Animal study

C57BL/6 WT mice were purchased from Hubei Research Center of Laboratory Animals (Wuhan, Hubei, China). C57BL/6 NLRP3$^{-/-}$ mice were kindly provided by Dr. Di Wang of Zhejiang University School of Medicine, China.

The primary mouse embryo fibroblasts (MEFs) were prepared from pregnancy mice of C57BL/6 WT mice and C57BL/6 NLRP3$^{-/-}$ mice in E14 and cultured in Dulbecco modified Eagle medium (DMEM) containing 10% heat-inactivated fetal bovine serum (FBS).

### Ethics statement

All animal studies were performed in accordance with the principles described by the Animal Welfare Act and the National Institutes of Health Guidelines for the care and use of laboratory animals in biomedical research. All procedures involving mice and experimental protocols were approved by the Institutional Animal Care and Use Committee (IACUC) of the College of Life Sciences, Wuhan University.

### Cell lines and cultures

African green monkey kidney epithelial (Vero) cells, human cervical carcinoma (Hela) cells, and human embryonic kidney 293T (HEK 293T) cells were purchased from American Type Culture Collection (ATCC) (Manassas, VA, USA). Human acute monocytic leukemia (THP-1) cells were gift from Dr. Jun Cui of State Key Laboratory of Biocontrol, School of Life Sciences, Sun Yat-Sen University, Guangzhou 510275, China. THP-1 cells were cultured in RPMI 1640 medium supplemented with 10% heat-inactivated fetal bovine serum (FBS), 100 U/ml penicillin, and 100 μg/ml streptomycin sulfate. Vero, Hela and HEK293T cells were cultured in Dulbecco modified Eagle medium (DMEM) purchased from Gibco (Grand Island, NY, USA) supplemented with 10% FBS, 100 U/ml penicillin and 100 μg/ml streptomycin sulfate. Vero, Hela, HEK293T and THP-1 cells were maintained in an incubator at 37°C in a humidified atmosphere of 5% $CO_2$.

### Reagents

Lipopolysaccharide (LPS), adenosine triphosphate (ATP), Endoplasmic Reticulum Isolation Kit (ER0100), phorbol-12-myristate-13-acetate (TPA) and dansylsarcosine piperidinium salt (DSS) were purchased from Sigma-Aldrich (St. Louis, MO, USA). RPMI 1640 and Dulbecco modified Eagle medium (DMEM) were obtained from Gibco (Grand Island, NY, USA). Nigericin and Ac-YVAD-cmk were obtained from InvivoGene Biotech Co., Ltd. (San Diego, CA, USA). Antibody against Flag (F3165), HA (H6908) and monoclonal mouse anti-GAPDH

(G9295) were purchased from Sigma (St. Louis, MO, USA). Monoclonal rabbit anti-NLRP3 (D2P5E), Ubiquitin mouse mAb (P4D1), monoclonal rabbit anti-K63-linkage Specific Polyubiquitin (D7A11), monoclonal rabbit anti-K48-linkage Specific Polyubiquitin (D9D5), monoclonal rabbit anti-STING(D2P2F), monoclonal rabbit anti-calnexin(C5C9), monoclonal rabbit anti-IL-1β (D3U3E), IL-1β mouse mAb (3A6) and monoclonal rabbit anti-Caspase-1 (catalog no. 2225) were purchased from Cell Signaling Technology (Beverly, MA, USA). Monoclonal mouse anti-ASC (sc-271054) and polyclonal rabbit anti-IL-1β (sc-7884) were purchased from Santa Cruz Biotechnology (Santa Cruz, CA, USA). Monoclonal mouse anti-NLRP3 (AG-20B-0014-C100) was purchased from Adipogen to detection endogenous NLRP3 in THP1 cells and primary MEFs. Lipofectamine 2000, ER-Tracker™ Blue-White DPX (E12353), normal rabbit IgG and normal mouse IgG were purchased from Invitrogen Corporation (Carlsbad, CA, USA).

## Viruses

Herpes simplex virus 1 (HSV-1) strain and Sendai virus (SeV) strain were gifts from Dr. Bo Zhong of Wuhan University. Zika virus (ZIKV) isolate z16006 (GenBank accession number KU955589.1) was used in this study. Vero cells were maintained at 37˚C in DMEM (GIBCO) supplemented with 10% heat-inactivated FBS with penicillin/streptomycin (GIBCO) (Grand Island, NY, USA). HSV-1 stocks were propagated in Vero cells for 36 h at 0.03 MOI. The infected cells were collected after three times of freezing and thawing in the infected cells and titrated by plaque assay in 12-well plates in Vero cells. The mock-infected was prepared at the same Vero cells without HSV-1 infection, but others are the same procedure.

## Plaque assay

Vero cells were cultured in a 12-well plate at a density of $2 \times 10^5$ cells per well, and infected with 100 μl serially diluted HSV-1 supernatant for 2 h. Then, the cells were washed by PBS and then immediately replenished with plaque medium supplemented with 1% carboxylmethylcelluose. The infected Vero cells were incubated for 2–3 days. After the incubation, plaque medium was removed and cells were fixed and stained with 4% formaldehyde and 0.5% crystal violet.

## THP-1 macrophages stimulation

THP-1 cells were differentiated into macrophages with 100 ng/ml phorbol-12-myristate-13-acetate (TPA) for 12 h, and cells were cultured for 24 h without TPA. Differentiated cells were then stimulated with HSV-1 infection, HSV120 transfection, DNA90 transfection, Sendai virus (SeV) infection, Zika virus (ZIKV) infection, or Nigericin treatment. Supernatants were collected for measurement of IL-1β by Enzyme-linked immunosorbent assay (ELISA). Cells were harvested for real-time PCR or immunoblot analysis.

## Plasmid construction

The cDNAs encoding human STING, NLRP3, ASC, pro-Casp-1, and IL-1β were obtained by reverse transcription of total RNA from TPA-differentiated THP-1 cells, followed by PCR using specific primers. The cDNAs were sub-cloned into pcDNA3.1(+) and pcagg-HA vector. The pcDNA3.1(+)-3×Flag vector was constructed from pcDNA3.1(+) vector through inserting the 3×Flag sequence between the NheI and HindIII site. Following are the primers used in this study. Flag-NLRP3: 5'-CGCGGATCCATGAAGATGGCAAGCACCCGC-3', 5'-CCGCTCGA

GCTACCAAGAAGGCTCAAAGAC-3'; Flag-ASC: 5'-CCGGAATTCATGGGGCGCGCGCG CGACGCCAT-3', 5'-CCGCTCGAGTCAGCTCCGCTCCAGGTCCTCCA-3'; Flag-Casp-1: 5'-CGCGGATCCATGGCCGACAAGGTCCTGAAG-3', 5'-CCGCTCGAGTTAATGTCCT GGGAAGAGGTA-3'; Flag-IL-1β: 5'-CCGGAATTCATGGCAGAAGTACCTGAGCTC-3', 5'-CCGCTCGAGTTAGGAAGACACAAATTGCAT-3'; Flag-STING: 5'-CCGGAATTCTAT GCCCCACTCCAGCCTGCA-3', 5'-CCGCTCGAGTCAAGAGAAATCCGTGCGGAG-3'. HA-NLRP3: 5'-TACGAGCTCATGAAGATGGCAAGCACCCGC-3', 5'-CCGCTCGAGCC AAGAAGGCTCAAAGACGAC-3'; HA-ASC: 5'-CCGGAATTCATGGGGCGCGCGCGCGA CGCC-3', 5'-CCGCTCGAGGCTCCGCTCCAGGTCCTCCAC-3'; HA-Casp-1: 5'-CCGGAA TTCATGGCCGACAAGGTCCTGAAG-3', 5'-CCGCTCGAGATGTCCTGGGAAGAGGTA GAA-3'.

The STING truncates was cloned into pcDNA3.1(+) and the PYRIN, NACHT, and LRR domain of NLRP3 protein was cloned into pcDNA3.1(+) and pcaggs-HA vector using specific primers, which are listed as follows. Flag-STING(1–160): 5'-CGCGGATCCATGCCCCACTC CAGCCTGCAT-3', 5'-CCGGAATTCTGCCAGCCCATGGGCCACGTT-3'; Flag-STING(1– 240): 5'-CGCGGATCCATGCCCCACTCCAGCCTGCAT-3', 5'-CCGGAATTCGTAAACCC GATCCTTGATGCC-3; Flag-STING(41–379): 5'-CGCGGATCCATGGAGCACACTCTCC GGTAC-3', 5'-CCGGAATTCTCAAGAGAAATCCGTGCGGAG-3'; Flag-STING(81–379): 5'-CGCGGATCCATGTACTGGAGGACTGTGCGG-3', 5'-CCGGAATTCTCAAGAGAAAT CCGTGCGGAG-3'; Flag-STING(111–379): 5'-CGCGGATCCATGGCGGTCGGCCCGCCC TTC-3', 5'-CCGGAATTCTCAAGAGAAATCCGTGCGGAG-3'; Flag-STING(151–379): 5'-C GCGGATCCATGAATTTCAACGTGGCCCAT-3', 5'-CCGGAATTCTCAAGAGAAATC CGTGCGGAG-3', Flag-STING(211–379): 5'-CGCGGATCCATGCTGGATAAACTGCCC CAG-3', 5'-CCGGAATTCTCAAGAGAAATCCGTGCGGAG-3'; HA-PYRIN: 5'-CCGGAA TTCTGAAGATGGCAAGCACCCGC-3', 5'-CCGCTCGAGTAAACCCATCCACTCCTCT TC-3'; HA-NACHT: 5'-CCGGAATTCATGCTGGAGTACCTTTCGAGA-3', HA-LRR: 5'-AT CGAGCTCATGTCTCAGCAAATCAGGCTG-3', 5'-CCGCTCGAGCCAAGAAGGCTCAA AGACGAC-3'.

## DNA oligonucleotides

The following oligonucleotide was used to stimulate cells: dsDNA90: 5'-TACAGATCTACTA GTGATCTATGACTGATCTGTACATGATCTACATACAGATCTACTAGTGATCTATGA CTGATCTGTACATGATCTACA-3'. HSV120: 5'-AGACGGTATATTTTTGCGTTATCACT GTCCCGGATTGGACACGGTCTTGTGGGATAG GCATGCCCAGAAGGCATATTGGGT TAACCCCTTTTTATTTGTGGCGGGTTTTTTGGAGGACTT-3'.

## Lentivirus production and infection

The targeting sequences of shRNAs for the human STING, NLRP3, ASC and Casp-1 were as follows: sh-STING: GCCCGGATTCGAACTTACAAT; sh-NLRP3: 5'-CAGGTTTGACTATC TGTTCT-3'; sh-ASC: 5'-GATGCGGAAGCTCTTCAGTTTCA-3'; sh-caspase-1: 5'-GTGAA GAGATCCTTCTGTA-3'. A PLKO.1 vector encoding shRNA for a negative control (Sigma-Aldrich, St. Louis, MO, USA) or a specific target molecule (Sigma-Aldrich) was transfected into HEK293T cells together with psPAX2 and pMD2.G with Lipofectamine 2000. Culture supernatants were harvested 36 and 60 h after transfection and then centrifuged at 2,200 rpm for 15 min. THP-1 cells were infected with the supernatants contain lentiviral particles in the presence of 4 μg/ml polybrene (Sigma). After 48 h of culture, cells were selected by 1.5 μg/ml puromycin (Sigma). Hela cells were selected by 2.5 μg/ml puromycin (Sigma). The results of each sh-RNA-targeted protein were detected by immunoblot analysis.

### Enzyme-linked immunosorbent assay (ELISA)

Concentrations of human IL-1β in culture supernatants were measured by ELISA kit (BD Biosciences, San Jose, CA, USA). The mouse IL-1β ELISA Kit was purchased from R&D.

### Activated caspase-1 and mature IL-1β measurement

Supernatant of the cultured cells was collected for 1 ml in the cryogenic vials (Corning). The supernatant was centrifuged at 12,000 rpm for 10 min each time by Amicon Ultra (UFC500308) from Millipore for protein concentrate. The concentrated supernatant was mixed with SDS loading buffer for western blotting analysis with antibodies for detection of activated caspase-1 (D5782 1:500; Cell Signaling) or mature IL-1β (Asp116 1:500; Cell Signaling). Adherent cells in each well were lysed with the lyses buffer described below, followed by immunoblot analysis to determine the cellular content of various protein.

### Western blot analysis

HEK293T whole-cell lysates were prepared by lysing cells with buffer (50 mM Tris-HCl, pH7.5, 300 mM NaCl, 1% Triton-X, 5 mM EDTA and 10% glycerol). The TPA-differentiated THP-1 cells lysates were prepared by lysing cells with buffer (50 mM Tris-HCl, pH7.5, 150 mM NaCl, 0.1% Nonidetp 40, 5 mM EDTA and 10% glycerol). Protein concentration was determined by Bradford assay (Bio-Rad, Hercules, CA, USA). Cultured cell lysates (30 μg) were electrophoresed in an 8–12% SDS-PAGE gel and transferred to a PVDF membrane (Millipore, MA, USA). PVDF membranes were blocked with 5% skim milk in phosphate buffered saline with 0.1% Tween 20 (PBST) before being incubated with the antibody. Protein band were detected using a Luminescent image Analyzer (Fujifilm LAS-4000).

### Co-immunoprecipitation assays

HEK293T whole-cell lysates were prepared by lysing cells with buffer (50 mM Tris-HCl, pH7.5, 300 mM NaCl, 1% Triton-X, 5 mM EDTA, and 10% glycerol). TPA-differentiated THP-1 cells lysates were prepared by lysing cells with buffer (50 mM Tris-HCl, pH7.5, 150 mM NaCl, 0.1% Nonidetp40, 5 mM EDTA, and 10% glycerol). Lysates were immunoprecipitated with control mouse immunoglobulin G (IgG) (Invitrogen) or anti-Flag antibody (Sigma, F3165) with Protein-G Sepharose (GE Healthcare, Milwaukee, WI, USA).

### Confocal microscopy

HEK293T cells and Hela cells were transfected with plasmids for 24–36 h. Cells were fixed in 4% paraformaldehyde at room temperature for 15 min. After being washed three times with PBS, permeabilized with PBS containing 0.1% Triton X-100 for 5 min, washed three times with PBS, and finally blocked with PBS containing 5% BSA for 1 h. The cells were then incubated with the monoclonal mouse anti-Flag antibody (F3165, Sigma) and Monoclonal rabbit anti-HA (H6908, Sigma) overnight at 4˚C, followed by incubation with FITC-conjugate donkey anti-mouse IgG (Abbkine) and Dylight 649-conjugate donkey anti-rabbit IgG (Abbkine) for 1 h. After washing three times, cells were incubated with DAPI solution for 5 min, and then washed three more times with PBS. Finally, the cells were analyzed using a confocal laser scanning microscope (Fluo View FV1000; Olympus, Tokyo, Japan).

### Real-time PCR

Total RNA was extracted with TRIzol reagent (Invitrogen), following the manufacturer's instructions. Real-time quantitative-PCR was performed using the Roche LC480 and SYBR

qRT-PCR kits (DBI Bio-science, Ludwigshafen, Germany) in a reaction mixture of 20 μl SYBR Green PCR master mix, 1 μl DNA diluted template, and RNase-free water to complete the 20 μl volume. Real-time PCR primers were designed by Primer Premier 5.0 and their sequences were as follows: HSV-1 ICP27 forward, 5'-GCATCCTTCGTGTTTGTCATT-3', HSV-1 ICP27 reverse, 5'-GCATCTTCTCTCCGACCCCG-3'. HSV-1 UL30 forward, 5'-CA TCACCGACCCGGAGAGGGAC-3', HSV-1 UL30 reverse, 5'-GGGCCAGGCGCTTGTT GGTGTA-3'. SeV P protein forward, 5'- CAAAAGTGAGGGCGAAGGAGAA-3', SeV P protein reverse, 5'- CGCCCAGATCCTGAGATACAGA-3'. ZIKV forward, 5'-GGTCAGCGTC CTCTCTAATAAACG-3', ZIKV reverse, 5'-GCACCCTAGTGTCCACTTTTTCC-3'. IL-1β forward, 5'-CACGATGCACCTGTACGATCA-3', IL-1β reverse, 5'-GTTGCTCCATATCC TGTCCCT-3'. GAPDH forward, 5'-AAGGCTGTGGGCAAGG-3', GAPDH reverse, 5'-TGG AGGAGTGGGTGTCG-3'. Mouse GAPDH forward, 5'-TGGCCTTCCGTGTTCCTAC-3', Mouse GAPDH reverse, 5'-GAGTTGCTGTTGAAGTCGCA-3'. Mouse IL-1β forward, 5'-GA AATGCCACCTTTTGACAGTG-3', Mouse IL-1β reverse, 5'-TGGATGCTCTCATCAGGA CAG-3'. Mouse IL-6 forward, 5'-ACAAAGCCAGAGTCCTTCAGA -3', Mice IL-6 reverse, 5'-TCCTTAGCCACTCCTTCTGT-3'. Mice TNF-α forward, 5'-ACTGAACTTCGGGGGTGA TCG-3', Mice TNF-α reverse, 5'-TCTTTGAGATCCATGCCGTTG-3'.

## ASC oligomerization detection

HEK293T cells were transfected with plasmids for 24–36 h. Cell lysates were centrifuged at 6000 rpm for 15 min. The supernatants of the lysates were mixed with SDS loading buffer for western blot analysis with antibody against ASC. The pellets of the lysates were washed with PBS for three times and cross-linked using fresh DSS (2 mM, Sigma) at 37°C for 30 min. The cross-linked pellets were then spanned down and the supernatant was mixed with SDS loading buffer for western blotting analysis.

## Statistical analyses

All experiments were reproducible and repeated at least three times with similar results. Parallel samples were analyzed for normal distribution using Kolmogorov-Smirnov tests. Abnormal values were eliminated using a follow-up Grubbs test. Levene's test for equality of variances was performed, which provided information for Student's t-tests to distinguish the equality of means. Means were illustrated using histograms with error bars representing the SD; a P value of <0.05 was considered statistically significant.

## Supporting information

**S1 Fig. Determination of the expression of IFI16 during HSV-1 infection in THP-1 macrophages.** TPA-differentiated THP-1 macrophages were treated with 2 μM Nigericin for 2 h, and infected with HSV-1 at MOI = 0.8 for 2, 4, 6 and 8 h. The expression of IFI16 during HSV-1 infection was determined by Western-blot analyses.
(TIFF)

**S2 Fig. The localization of NLRP3 in Golgi after HSV-1 infection.** (**A**) Hela cells were transfected with pFlag-NLRP3 and infected with HSV-1 (MOI = 1) for 4 h or transfected with HSV120 (3 μg/ml) for 4 h. Sub-cellular localization of Flag-NLRP3 (green), Syntaxin 6 (TGN marker, red) and DAPI (blue) were examined by confocal microscopy. (**B**) Hela cells were transfected with pHA-NLRP3 and infected with HSV-1 (MOI = 1) for 4 h or transfected with HSV120 (3 μg/ml) for 4 h. Sub-cellular localization of HA-NLRP3 (red), GM130 (cis Golgi marker, green) and DAPI (blue) were examined by confocal microscopy. (**C**) TPA-

differentiated THP-1 macrophages were infected with mock or HSV-1 (MOI = 1) for 4 h. Subcellular localization of NLRP3 (green), Syntaxin 6 (TGN marker, red) and DAPI (blue) were examined by confocal microscopy.
(TIFF)

## Acknowledgments

We thank Dr. Di Wang of Zhejiang University School of Medicine, China for kindly providing C57BL/6 NLRP3$^{-/-}$ mice, Dr. Jun Cui of Sun Yat-Sen University, Guangzhou 510275, China for Human acute monocytic leukemia (THP-1) cells, and Dr. Bo Zhong of Wuhan University, China, for kindly providing Herpes simplex virus 1 strain, Sendai virus strain, and plasmids encoding ubiquitin and mutants.

## Author Contributions

**Conceptualization:** Wenbiao Wang, Kailang Wu, Jianguo Wu.

**Data curation:** Wenbiao Wang, Dingwen Hu, Caifeng Wu, Yuqian Feng, Aixin Li, Weiyong Liu, Yingchong Wang, Keli Chen, Mingfu Tian, Feng Xiao, Qi Zhang, Muhammad Adnan Shereen, Weijie Chen, Pan Pan, Pin Wan, Jianguo Wu.

**Formal analysis:** Wenbiao Wang, Dingwen Hu, Weiyong Liu, Yingchong Wang, Keli Chen, Mingfu Tian, Feng Xiao, Qi Zhang, Muhammad Adnan Shereen, Weijie Chen, Pan Pan, Pin Wan.

**Funding acquisition:** Kailang Wu, Jianguo Wu.

**Investigation:** Wenbiao Wang, Dingwen Hu, Caifeng Wu, Yuqian Feng, Aixin Li, Weiyong Liu, Yingchong Wang, Keli Chen, Mingfu Tian, Feng Xiao, Qi Zhang, Muhammad Adnan Shereen, Weijie Chen, Pan Pan, Pin Wan, Kailang Wu, Jianguo Wu.

**Methodology:** Wenbiao Wang, Dingwen Hu, Caifeng Wu, Yuqian Feng, Aixin Li, Weiyong Liu, Yingchong Wang, Keli Chen, Mingfu Tian, Feng Xiao, Qi Zhang, Muhammad Adnan Shereen, Weijie Chen, Pan Pan, Pin Wan.

**Resources:** Wenbiao Wang, Aixin Li, Weiyong Liu, Keli Chen, Mingfu Tian, Feng Xiao, Kailang Wu.

**Supervision:** Kailang Wu, Jianguo Wu.

**Validation:** Wenbiao Wang, Dingwen Hu, Caifeng Wu, Yuqian Feng, Aixin Li, Kailang Wu, Jianguo Wu.

**Visualization:** Wenbiao Wang, Dingwen Hu, Caifeng Wu, Aixin Li, Kailang Wu.

**Writing – original draft:** Wenbiao Wang, Jianguo Wu.

**Writing – review & editing:** Jianguo Wu.

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
