## [Decision Letter · Decision Letter 0]

28 Sep 2019

Dear Dr. Wu,

Thank you very much for submitting your manuscript "STING recruits NLRP3 to the ER and deubiquitinates NLRP3 to activate the inflammasome upon DNA virus infection" (PPATHOGENS-D-19-01557) for review by PLOS Pathogens. Your manuscript was fully evaluated at the editorial level and by independent peer reviewers. The reviewers appreciated the attention to an important problem, but raised some substantial concerns about the manuscript as it currently stands. These issues must be addressed before we would be willing to consider a revised version of your study. We cannot, of course, promise publication at that time.

We therefore ask you to modify the manuscript according to the review recommendations before we can consider your manuscript for acceptance. Your revisions should address the specific points made by each reviewer. In particular, the roles of cGAS and AIM/IFI16 should be addressed. Please also pay attention to the experimental procedures, for example, the preparation of viral inoculum. In order to justify the title, additional families of DNA viruses eg vaccinia virus should be included.

(1) A letter containing a detailed list of your responses to the review comments and a description of the changes you have made in the manuscript. Please note while forming your response, if your article is accepted, you may have the opportunity to make the peer review history publicly available. The record will include editor decision letters (with reviews) and your responses to reviewer comments. If eligible, we will contact you to opt in or out.

(2) Two versions of the manuscript: one with either highlights or tracked changes denoting where the text has been changed; the other a clean version (uploaded as the manuscript file).

Additionally, to enhance the reproducibility of your results, PLOS recommends that you deposit your laboratory protocols in protocols.io, where a protocol can be assigned its own identifier (DOI) such that it can be cited independently in the future. For instructions see http://journals.plos.org/plospathogens/s/submission-guidelines#loc-materials-and-methods

We hope to receive your revised manuscript within 60 days. If you anticipate any delay in its return, we ask that you let us know the expected resubmission date by replying to this email. Revised manuscripts received beyond 60 days may require evaluation and peer review similar to that applied to newly submitted manuscripts.

[LINK]

Sincerely,

Fanxiu Zhu, Ph.D.

Associate Editor

PLOS Pathogens

Shou-Jiang Gao

Section Editor

PLOS Pathogens

Kasturi Haldar

Editor-in-Chief

PLOS Pathogens

orcid.org/0000-0001-5065-158X

Grant McFadden

Editor-in-Chief

PLOS Pathogens

orcid.org/0000-0002-2556-3526

Reviewer's Responses to Questions

**Part I - Summary**

Reviewer #1: In this paper, Wu and coworkers suggested the requirement of cGAS-STING pathway of DNA sensing for activation of NLRP3 inflammasome during HSV-1 infection. It was previously hypothesized that HSV-1 induces NLRP3 inflammasome through an unknown mechanism. This work provides one possible explanation and some new insight on how host activates NLRP3 inflammasome. However, while some results are interesting, others are redundant and confirmatory. In addition, several key questions remain unanswered. Some parts are disconnected. There are major concerns that have to be addressed. Specific comments are as follows.

Reviewer #2: Manuscript: Wenbiao Wang et al: STING recruits NLRP3 to the ER and deubiquitinates NLRP3 to activate the inflammasome upon DNA virus infection

Background: Inflammasome induction and type I The DNA Inflammasome are the first line of defense in eukaryotic cells against viruses etc. The presence of cytosolic DNA is recognized by several pattern receptors- the cGAS-STING axis triggers type I interferon response and the NLRP3 and AIM2 axis triggers the inflammasome axis. NLRP3 is activated by a broad range of pathogen-derived and endogenous agents including mitochondrial DNA etc.

Recent studies reveal the complexity of these stimulation, cell type differences, and the cross-interaction and stimulation among the components. For example:

A. Feb 2017 Nature communications: Studies demonstrate that “ IFI16 and cGAS cooperate in the activation of STING during DNA sensing in human keratinocytes”

B. Cell 2017 studies demonstrate that “DNA in Human Myeloid Cells Is Initiated by a STING-Cell Death Program Upstream of NLRP3”

Cell 2017 studies show that cell type-specificities and differences between primary monocytes and THP-1 cells.

1) In human monocytic cell line THP-1 and murine BMDMs (bone marrow derived macrophages), AIM2 mediates the cytosolic DNA recognition, inflammasome activation and IL-1b secretion, and not by NLRP3 in THP-1 cells, and pharmacologic inhibition of NLRP3 had no impact on DNA-mediated inflammasome activation in these cells

2) In contrast, NLRP3 is involved in the inflammasome induction after cytosolic DNA recognition in human monocytes from peripheral blood.

C. J Expt medicine 2017 paper show that cGAMP has a noncanonical function in inflammasome activation in human and mouse cells.

In BMDMs, cGAMP activates the inflammasome through an AIM2, NLRP3, ASC, and caspase-1 dependent process, requiring differential requirements for STING. cGAMP induction of IFN-I preceded the inflammasome activation, and both cGAS/cGAMP amplify both inflammasome and IFN-I responses to control murine cytomegalovirus in mice.

D. HSV-1 tegument protein VP22 of HSV-1 has been shown to inhibit AIM2 inflammasome induction.

E. Human fibroblast cell infection induces the NLRP3-ASC interaction o-4 h infection but not later.

Major:

This manuscript follows some of these papers to determine the induction of NLRP3 inflammasome induction by HSV-1.

While some findings are very interesting, this manuscript is written in a brief format including abstract, introduction, results and discussion section. A difficult read and several experimental details are lacking, and hence difficult to follow, lacks appropriate controls and very selective. See the comments under individual figures.

The overall conclusion Fig 9- Role of cGAS in viral DNA recognition and activation of STING is not shown in any of the figures given.

Title: STING recruits NLRP3 to the ER and deubiquitinates NLRP3 to activate the inflammasome upon DNA virus infection

“ upon DNA virus infection” – They have tested only HSV-1. If they have also tested cytoplasmic replicating vaccinia virus as a control – it will be valid- as vaccinia virus has been shown to stimulate AIM2 in human cells.

Reviewer #3: This manuscript describes that DNA virus activates NLRP3 inflammasome through STING. It shows NLRP3 inflammasome could be activated by viral DNAs through STING pathway in addition to the well-known ROS and other PAMPs, STING-mediated NLRP3 inflammasome activation is mainly through recruiting NLRP3 to the ER and removing K48- and K63-linked polyubiquitination of NLRP3. This manuscript is quite interesting that it extends the range of NLRP3 activation and offers a novel perspective of STING’s function in unification of innate immunity and inflammasome.

Reviewer #4: In this study, the authors reveal a distinct function of STING of regulating NLRP3 dependent inflammasome activation in multiple cells/cell lines upon HSV-1 infection or cytosolic DNA stimulation. Mechanistically, the authors provide strong evidence that STING binds to NLRP3 in ER and therefore stabilizing the inflammasome formation. Furthermore, they also showed that STING overexpression is associated with attenuation of both K48- and K63-linked polyubiquitination of NLRP3, thereby promoting the inflammasome activation. This is a very interesting and solid study with decent amounts of supporting data. I would like to recommend that the authors perform some extra experiments, discuss more about the mechanism and make more appropriate conclusions.

**Part II – Major Issues: Key Experiments Required for Acceptance**

Reviewer #1: 1) Many key questions that should be addressed and can be easily addressed were left untouched. These include the following and at least some should be experimentally addressed: 

i) The role of cGAS: It was mentioned in the abstract that cGAS-STING-NLRP3 pathway is essential for host defense against DNA virus infection. However, no experiment was done to build the link with cGAS. 

ii) The role of cGAMP: Is STING activated by cGAMP or directly by DNA or anything else?

iii) The identity of the deubiquitinase recruited.

iv) iv) Does STING per se have deubiquitinase activity towards NLRP3?

2)   Why were LPS and nigericin used to prime cells (Fig. 3, 4) if HSV-1 alone can sufficiently activate both signals for inflammasome activation (Fig. 7, 8).

Reviewer #2: Fig.1. HEK293 T cells were used (A-L). These cells lack several innate response proteins most notable cGAS, IFI16 etc. (Proc Natl Acad Sci U S A. 2015 Apr 7; 112(14): E1773–E1781.

- Fig 9- Steady state – STING and NLRP3 do not interact- yet they show over expression of these molecules in 293T and HeLa they interact? How?

- Appropriate controls are lacking.

- Why STING and NLRP3 interacts in the absence of cGAS or cGAMP and any external stimulation? How inflammasome is induced in the absence of recognition of DNA? Or other stimulus? Not explained and very difficult to conclude without the experimental details

- Show cGAS?

Line 116- 117 – Explain the system.

Fig 2:

-Appropriate controls lacking.

Infection procedures are not clear. What is 2 h/ is it 2 h incubation with virus, wash and then 2 and 4 h infection?

- What is Mock? Not described.

- Lines 434-436: HSV-1 was collected by freezing and thawing infected Vero cells- Hence, this is not a purified enveloped virus preparation.This preparation with contain enveloped viruses, viral DNA, host nuclear and mitochondrial DNA, viral capsids with and without daughter DNA, cell debris. All these materials added to the cell can be taken up by endocytosis and can induce the stimulus.

- TPA induced THP1 cells were used for HSV-1 infection (2-4 h) - AIM2 has been shown to induce the inflammasome after recognition of the cytoplasmic DNA in these cells – see the background above.

- No AIM2 controls.

- Fig 2A. Fig 1 shows interaction of NLRP3 and STING- here no endogenous interaction- explain. But there is interaction of transfected DNA- C and D – lanes 2.

- HSV-1 is a nuclear replicating virus and IFI16-inflammasome and interferon responses have been shown to be stimulated- Show this here as controls.

- What about cGAS and cGAMP production in the endogenous infection conditions?

- Fig 2E: Control plasmid transfection?

- So why transfection of HSV120 plasmid is not inducing AIM2 inflammasome? This lacks VP22. Hence AIM2 inflammasome will be induced.

This DNA has to enter the nucleus in order to express the viral genes (as shown in the later figures) and hence must be inducing IFI16-cGAS interferon response and IFI16-inflammasome.

Hence, the results as shown are very surprising.

Fig.3:

HSV-1 infection also induces NLRP3 and IFI16 inflammasome at early time of infection (up to 4h) and IFI16 is degraded after 2-4 h in the fibroblasts (Ref 24).

To differentiate the responses that they observed in THP-1 cells, they should show the fate of IFI16.

Line 165: “ Induced upon DNA virus infection and cytosolic DNA stimulation.- Not correct statement. Vaccinia virus control should be included.

Fig.4.

Need to show time kinetics of HSV-1 infection induced inflammasome activation – ie Il1 beta and caspase 1 cleavage. in THP1 cells and mouse cells .

- Controls for ShRNA? Show AIM2 KD.

- Needs AIM2 control in Fig L, M and N

Fig. 5:

-Show AIM2 and IFI16 controls.

-STING and NLRP3 interactions? Does it require cGAS- Not clear and need to be shown.

- Show the endogenous levels of the molecules in the Hela cells.

- Since HSV120 is biotinlated DNA- should show the colocalization of NLRP3 and cGAS

- L: Primary MEFs stably expressing sh-STNG- this is surprising- Methods?

- STING Doesn’t appear to be efficiently reduced.

- LPS primed and infected? Why? Not clear.

Fig. 7:

- What is the effect of STING KD on type I interferon response and HSV-1 replication in these cells?

Line 315: “ This study reveals a distinct mechanism by which the cGAS-STING-NLRP3 pathway promotes the NLRP3 inflammasome activation and IL-1β secretion upon DNA virus infection and cytosolic DNA stimulation”

Fig. 9: Role of cGAS in viral DNA recognition and activation of STING is not shown in any of the figures given.

Reviewer #3: However, although potentially important and meaningful, there are several points need to be improved greatly, as outlined below, to make the conclusion more convincing and clear.

Major points:

1. In Figure 1, the author proved that NLRP3 interacts with STING, and NAD and LRR in NLRP3 play the main role, and TM2 and TM5 in STING are essential. Could the author map the smaller domains or the key residues for their interaction?

2. In Figure 1H, the expression of truncated proteins is not consistent, especially STING (211-379aa), thus the conclusion that TM5 of STING interacts with NLRP3 is not convincing. Furthermore, the plasmid expressing only TM5 or TM5-deleted STING plasmid should be added in this figure.

3. In Figure 1I, TM2 was claimed to promote IL-1β secretion. To be more convincing, this figure needs more mutant STING plasmids such as TM2, TM2-deleted, TM2-TM5 and TM2-TM5-deleted constructs.

4. HSV as an AIM2 inflammasome stimuli is well known for a long time. In Figure 4A, 4C, IL-1β secretion could hardly be recognized as the result of HSV stimulation. Further, it is VP22 but not intact HSV virion inhibits AIM2-dependent inflammasome according to Johnson’s paper, so it may be important to perform this experiment in AIM2 knock out THP-1 cells to identify if IL-1β would be still expressed in the same pattern as in THP-1 WT cells.

5. STING recruits NLRP3 to the ER, but from Figure 5A, it shows that NLRP3 is partially locating in ER without STING’s recruitment. In Figure 5B and 5C Calnexin and ER blue are performed to show the colocalization of ER and NLRP3, Zhijian Chen’s paper shows that NLRP3 aggregates in TGN under multiple stimulation such as Nigericin and ATP, could the authors stain TGN38/GM130 to determinate that NLRP3 is not locating in Golgi after HSV stimulation or HSV-120 transfection as well as stain EEA1 which could exclude the possibility that NLRP3 is not aggregated in endosome? In Figure 5J, 5K and 5L, NLRP3 still significantly augment in ER after HSV stimulation under the condition of STING knock down, it is also important to use STING-/- cells to identify if STING deficiency could affect NLRP3’s translocation into ER.

6. In Figure 6, the author should explain why HA-UB (K48R) and HA-UB (K63R) constructs were used. In Figure 6B-C, the difference and function of K48R, K48O, K63R and K63O should be described in main text.

7. In Figure 8, the authors have used NLRP3-KO mice, however, STING-KO mice may be more important in this study to prove its essential role in NLRP3 inflammasome activation. In Figure 8C, a RNA virus, such as SeV or ZIKA, should be added as a control.

8. In Figure 8, the author only described the important role of NLRP3 in host defense against viral infection. But whether the effect is related to the STING-NLRP3 association is not described. More experiments or description need to be added.

Reviewer #4: 1. Based on Figure 1H and 1I, STING 1-160 (transmembrane domain) is responsible for the NLRP3 binding as well as NLRP3 function, can the authors show that STING 1-160 can recruit NLRP3 that same as STING WT, like in Figure 1L-M?

2. Figure 5, upon HSV-1 infection as well DNA stimulation, STING has been reported to translocate to Golgi. Can the authors look further details on the location of STING regarding Golgi location? Moreover, in Figure 5H and 5I, it will be great to include STING IF staining.

3. Figure 8, it will be great if the authors can compare the IL-1b response in WT and STING-/- mice. (Do not have to do the survival curve, just show the IL-1b response in more than one organs like in 8C,D,H, K)

**Part III – Minor Issues: Editorial and Data Presentation Modifications**

Reviewer #1: 5)      Fig. 1C: ASC and pro-CASP1 were expressed in lower level. Will it be expression problem so that only NLRP3 can be detected? Reciprocal IP on STING should be performed to address this problem.

Fig. 1F, lanes 5-8: It is difficult to conclude that LRR and NACHT were immunoprecipitated with STING. A better blot or labelling is required to demonstrate the corresponding bands of LRR and NACHT clearly. Secondary antibodies that react strongly with the heavy chain of IgG should be avoided.

Fig. 1G, H and I: Apparently TM4 of STING is also involved in the interaction with NLRP3. The pull down of NLRP3 was highly reduced in lane 7. A truncated mutant with only TM4 and TM5 removed (1-111, 160-379bp) should be constructed and its function on NLRP3 binding and activation should be verified. This should also be included in ubiquitination assays and other experiments.

STING interacts with NLRP3 through TM5 domain (Fig. 1H) and loss of TM2 abolishes STING-mediated IL-1β induction (Fig 1I). Experiment presented in Fig. 1J, K should be repeated using STING binding-defective mutant (delTM5) and delTM3 to make the point that both TMs are necessary for STING-mediated NLRP3 activation. Truncated mutation of STING (delTM2 and delTM5) can also be used to strengthen the point that STING recruits and deubiquitinates NLRP3 upon HSV-1 infection (Fig. 5, 6).  

Fig. 1J: The label of input is missing in the lower panel.

Fig. 1K: Oligomer was detected in negative control (ASC only).

6)      Fig. 2B: STING was also pulled down by negative control IgG.

Fig. 2G: The IP label for the first lane should be Flag instead of IgG. STING can be pulled down in the first lane. The label should be similar to Fig. 2D.

7)      Fig. 5: Staining and fractionation assays were used to suggest that STING recruits NLRP3 to ER for inflammasome activation. NLRP3 and ASC speck formation should be analyzed to provide better evidence in support of the claim (Fig. 5H and 5I).

STING can be translocated into ERGIC and Golgi upon activation, to clarify whether NLRP3 is also recruited, Golgi and EGRIC markers should be used upon HSV-1 infection and HSV120-mer transfection. Moreover, it was previously reported that NLRP3 can be translocated to mitochondria through MAVS. Thus, NLRP3 should be costained with mitochondrial marker upon HSV-1 infection.

For the fractionation assay (Fig. 5E, F, G, J, K, L), it is possible that the microsomal fraction might be contaminated by Golgi and EGRIC. Since it was not explained how fractionation was performed, Golgi and EGRIC markers should be analyzed by Western blotting.

In loss-of-function experiment, it was shown that knockdown of STING prevents NLRP3 translocation to ER (Fig. 5) and its deubiquitination (Fig. 6). The critical experiment here is the detection of IL-1β protein level (Fig. 7). Additional lines of results are required to address whether knockdown of STING reduces ASC speck formation or oligomerization upon HSV-1 infection. STING inhibitors can also be used to test this hypothesis. 

8)      Fig. 6: Control of nigericin treated sample should be added to show the requirement of STING for HSV-1-induced NLRP3 inflammasome only.

9)      Fig. 8N, O: H&E staining was shown for pathology of HSV-1-infected WT and NLRP3-/- mice. Tissue section of mock-infected WT and NLRP3-/- mice should also be shown. 

In text related to Fig. 8N and O. “Hematoxylin and Eosin (H&E) staining showed that more infiltrated neutrophils and mononuclear cells were detected in the lung and brain of infected WT mice as compared with NLRP3-/- mice (Fig 8N), and immunohistochemistry (IHC) analysis revealed that IL-1β protein level was higher in the lung and brain of infected WT mice as compared with NLRP3-/- mice (Fig 8O), revealing that NLRP3 deficiency mice are more susceptibility to HSV-1 infection and elicit weak inflammatory responses.” It should be mouse lung in Fig. 8N and mouse brain in Fig. 8O, not H&E in Fig. 8N and IHC in Fig. 8O.

Reviewer #2: Minor:

1. Introduction: Line 83-92: Very brief. Should cite the articles about HSV1 innate inflammasome and interferon responses and discuss the previous findings.

2. Transfection procedures are not given

3. Line 147-148. Give reference.

4. Methodologies are not given for many experiments.

5. Fig 9 legend- rewrite- line 874-876. Genomic DNA recognition by cGAS is not shown

Reviewer #3: Minor points:

1. In this article, the author may need to show the pro-IL-1β and p17, pro-CASP-1 and p20 in one figure instead of cutting into pro- and mature forms.

2. Some references are inaccurate, for example, to describe “The NLRP3 inflammasome is required for HSV-1-induced IL-1β activation”, authors have cited an article “Herpes Simplex Virus 1 VP22 Inhibits AIM2-Dependent Inflammasome Activation to Enable Efficient Viral Replication [25]”.

3. In Figure 1A-C, there are two different controls (IgG or HA-vector) that the authors used in co-immunoprecipitation experiments. In consideration of over-expressed proteins rather than endogenous proteins in HEK293T cells, HA-vector is more suitable control.

4. In line 270, L-1β should be IL-1β.

Reviewer #4: 1. MEF stands for “Mouse Embryonic Fibroblasts”, so “Mice” in front of “MEF” is redundant. Please note that there are multiple incidences in this manuscripts.

2. I do not think sufficient data has been provided to conclude that STING “removes” K48- and K63-linked polyubiquitination of NLRP3, thereby promoting the inflammasome activation. There is also no published articles indicating that STING has DUB function. The authors instead showed that STING expression is correlated with the attenuation of both K48- and K63-linked polyubiquitination of NLRP3. Would the authors consider concluding the results in a different way?

3. Line 353-364, would the authors please discuss possible mechanisms why STING regulates polyubiquitination of NLRP3? K63 linked polyubiquitination of NLRP3 has been shown to be critical for NLRP3 signaling. Would the authors discuss why removing this would benefit HSV-1 dependent NLRP3 activation?

PLOS authors have the option to publish the peer review history of their article (what does this mean?). If published, this will include your full peer review and any attached files.

Reviewer #1: No

Reviewer #2: No

Reviewer #3: No

Reviewer #4: No

---

## [Decision Letter · Decision Letter 1]

13 Jan 2020

Dear Dr. Wu:

Thank you very much for submitting your manuscript "STING promotes NLRP3 localization in ER and facilitates NLRP3 deubiquitination to activate the inflammasome upon HSV-1 infection" (PPATHOGENS-D-19-01557R1) for review by PLOS Pathogens. Your manuscript was fully evaluated at the editorial level and by independent peer reviewers. The reviewers appreciated the attention to an important topic but identified some aspects of the manuscript that should be improved.

We therefore ask you to modify the manuscript according to the review recommendations before we can consider your manuscript for acceptance. Your revisions should address the specific points made by each reviewer.

(1) A letter containing a detailed list of your responses to the review comments and a description of the changes you have made in the manuscript. Please note while forming your response, if your article is accepted, you may have the opportunity to make the peer review history publicly available. The record will include editor decision letters (with reviews) and your responses to reviewer comments. If eligible, we will contact you to opt in or out.

(2) Two versions of the manuscript: one with either highlights or tracked changes denoting where the text has been changed; the other a clean version (uploaded as the manuscript file).

We hope to receive your revised manuscript within 60 days or less. If you anticipate any delay in its return, we ask that you let us know the expected resubmission date by replying to this email.

[LINK]

Sincerely,

Fanxiu Zhu, Ph.D.

Associate Editor

PLOS Pathogens

Shou-Jiang Gao

Section Editor

PLOS Pathogens

Kasturi Haldar

Editor-in-Chief

PLOS Pathogens

orcid.org/0000-0001-5065-158X

Michael Malim

Editor-in-Chief

PLOS Pathogens

orcid.org/0000-0002-7699-2064

Reviewer's Responses to Questions

**Part I - Summary**

Reviewer #1: Most concerns raised have been addressed.

Reviewer #3: This manuscript has been greatly improved, and I am satisfied for the responses.

Reviewer #4: The authors made significant progress and adequately addressed my comments scientifically by either experiments or discussion. I would recommend acceptance. Well done and look forward to the authors' future work.

**Part II – Major Issues: Key Experiments Required for Acceptance**

Reviewer #1: Nil.

Reviewer #3: No more experiment is required.

Reviewer #4: The authors made significant progress in the last revision and adequately addressed my comments scientifically by either experiments or discussion.

**Part III – Minor Issues: Editorial and Data Presentation Modifications**

Reviewer #1: 1) The pixel and resolution of some Western blots are low, e.g. Fig. 3C and D Sup, Fig. 6F and I and Fig. 7G. Higher resolution blots should be provided for publication. In addition, densitometry should be performed for some of the blots if comparison is made, e.g. Fig. 2F, 3C, 3D, 3J, 4B, 4E, 4I, 4J, 5F-H, 5K-M, 7B, 7E, 7G and 7K.

2) For all confocal images, it would be desirable to show more than one cell (Fig. 1N, 5A, 5B, 5D, 5I, 5J).

Reviewer #3: (No Response)

Reviewer #4: The authors made significant progress in the last revision and adequately addressed my comments scientifically by either experiments or discussion.

PLOS authors have the option to publish the peer review history of their article (what does this mean?). If published, this will include your full peer review and any attached files.

Reviewer #1: No

Reviewer #3: No

Reviewer #4: Yes: Zhe Ma

---

## [Editor Report · Decision Letter 2]

19 Jan 2020

Dear Dr. Wu,

We are pleased to inform you that your manuscript 'STING promotes NLRP3 localization in ER and facilitates NLRP3 deubiquitination to activate the inflammasome upon HSV-1 infection' has been provisionally accepted for publication in PLOS Pathogens.

Before your manuscript can be formally accepted you will need to complete some formatting changes, which you will receive in a follow up email. A member of our team will be in touch within two working days with a set of requests.

Best regards,

Fanxiu Zhu, Ph.D.

Associate Editor

PLOS Pathogens

Shou-Jiang Gao

Section Editor

PLOS Pathogens

Kasturi Haldar

Editor-in-Chief

PLOS Pathogens

orcid.org/0000-0001-5065-158X

Michael Malim

Editor-in-Chief

PLOS Pathogens

orcid.org/0000-0002-7699-2064
---

## [Editor Report · Acceptance letter]

4 Mar 2020

Dear Dr. Wu,

We are delighted to inform you that your manuscript, "STING promotes NLRP3 localization in ER and facilitates NLRP3 deubiquitination to activate the inflammasome upon HSV-1 infection," has been formally accepted for publication in PLOS Pathogens.

Best regards,

Kasturi Haldar

Editor-in-Chief

PLOS Pathogens

orcid.org/0000-0001-5065-158X

Michael Malim

Editor-in-Chief

PLOS Pathogens

orcid.org/0000-0002-7699-2064